# VECTOR GRIMOIRE: Codebook-based Shape Generation under Raster Image Supervision

**Marco Cipriano** [* 1]  **Moritz Feuerpfeil** [* 1]  **Gerard de Melo** [1]

## Abstract

Scalable Vector Graphics (SVG) is a popular format on the web and in the design industry. However, despite the great strides made in generative modeling, SVG has remained underexplored due to the discrete and complex nature of such data. We introduce GRIMOIRE, a text-guided SVG generative model that is comprised of two modules: A Visual Shape Quantizer (VSQ) learns to map raster images onto a discrete codebook by reconstructing them as vector shapes, and an Auto-Regressive Transformer (ART) models the joint probability distribution over shape tokens, positions, and textual descriptions, allowing us to generate vector graphics from natural language. Unlike existing models that require direct supervision from SVG data, GRIMOIRE learns shape image patches using only raster image supervision which opens up vector generative modeling to significantly more data. We demonstrate the effectiveness of our method by fitting GRIMOIRE for closed filled shapes on MNIST and Emoji, and for outline strokes on icon and font data, surpassing previous image-supervised methods in generative quality and the vector-supervised approach in flexibility.

## 1. Introduction

In the domain of computer graphics, Scalable Vector Graphics (SVG) has emerged as a versatile format, enabling the representation of 2D graphics with precision and scalability. SVG is an XML-based vector graphics format that describes a series of parametrized shape primitives rather than a limited-resolution raster of pixel values. While modern generative models have made significant advancements in producing high-quality raster images (Ho et al., 2020;

---
[*]Equal contribution [1]Hasso Plattner Institute. Correspondence to: Marco Cipriano <marco.cipriano@hpi.de>.

*Proceedings of the 42nd International Conference on Machine Learning*, Vancouver, Canada. PMLR 267, 2025. Copyright 2025 by the author(s).

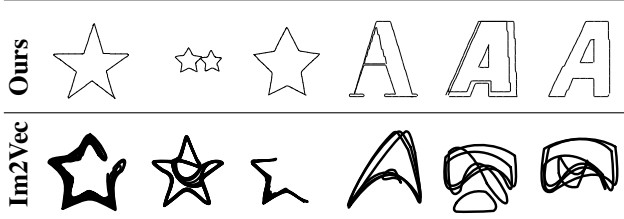

Figure 1. Generative results for fonts and icons from GRIMOIRE and Im2Vec. Since Im2Vec does not accept any conditioning, we sample after training Im2Vec only on icons of stars or the letter A, respectively. For GRIMOIRE we use the models trained on the full dataset conditioned on the respective class.

Isola et al., 2017; Saharia et al., 2022; Nichol et al., 2021), SVG generation remains a less explored task. Prior work aimed at training a deep neural network for this goal has primarily adopted large language models (LLMs) to address the problem (Wu et al., 2023; Tang et al., 2024). In general, existing approaches share two key limitations: they necessitate SVG data for direct supervision, which inherently limits the available data and increases the burden of data pre-processing, and they are not easily extendable when it comes to visual attributes such as color or stroke properties. The extensive pre-processing is required due to the diverse nature of an SVG file that can express shapes as a series of different basic primitives such as circles, lines, and squares – each having different properties – that can overlap and occlude each other.

An ideal generative model for SVG should however benefit from *visual guidance for supervision*, which is not possible when merely training to reproduce tokenized SVG primitives, as there is no differentiable mapping to the generated raster imagery. In this paper, we present GRIMOIRE, a novel pipeline explicitly designed to generate SVG files with only raster image supervision. Our approach incorporates a differentiable rasterizer, DiffVG (Li et al., 2020), to bridge the vector graphics primitives and the raster image domain. We adopt a VQ-VAE recipe (Van Den Oord et al., 2017), which pairs a codebook-based discrete auto-encoder with an auto-regressive Transformer that models the image space implicitly by learning the distribution of codes that resemble them. We find this approach particularly promising for

vector graphics generation, as it breaks the complexity of this task into two stages. In the first stage of our method, we decompose images into primitive shapes represented as patches. A vector-quantized auto-encoder learns to encode and map each patch into a discrete codebook, and decode these codes to an SVG approximation of the input patch, which is trained under raster supervision. In the second stage, the series of raster patches containing primitives are encoded and the prior distribution of codes is learned by an auto-regressive Transformer model conditioned on a textual description. At inference, a full series of codes can be generated from textual input, or other existing shape codes. Therefore, GRIMOIRE supports text-to-SVG generation and SVG auto-completion as possible downstream tasks out-of-the-box.

The key contributions of this work are:

1. We frame the problem of image-supervised SVG generation as the prediction of a series of individual shapes and their positions on a shared canvas.

2. We train the first text-conditioned generative model that learns to draw vector graphics with only raster image supervision.

3. We compare our model with alternative frameworks showing superior performance in generative capabilities on diverse datasets.

4. We release the code of this work to the research community[1].

## 2. Related Work

### 2.1. SVG Generative Models

The field of vector graphics generation has witnessed increasing interest. While a number of works predate the era of Large Language Models (Carlier et al., 2020; Wang & Lian, 2021), the most recent approaches (Lopes et al., 2019; Aoki & Aizawa, 2022; Wu et al., 2023; Tang et al., 2024) have recast the problem as an NLP task, learning a distribution over tokenized SVG commands. Iconshop (Wu et al., 2023) introduced a method of tokenizing SVG paths that makes them suitable input for causal language modeling. To add conditioning, they employed a pre-trained language model to tokenize and embed textual descriptions, which are concatenated with the SVG tokens to form sequences, on which the auto-regressive Transformer can learn a joint probability. Chat2SVG introduces a hybrid framework that leverages LLMs and image diffusion models to generate and refine SVG (Wu et al., 2024). StarVector trains a multimodal LLM on SVG data (Rodriguez et al., 2024). Stro-

keNUWA (Tang et al., 2024) introduced Vector Quantized Strokes to compress SVG strokes into a codebook with SVG supervision and fine-tune a pre-trained Encoder–Decoder LLM to predict these tokens given textual input.

Another line of work has sought to incorporate visual supervision. These approaches generally rely on recent advances in differentiable rasterization, which enables backpropagation of raster-based losses through different types of vectorial primitives such as Bézier curves, circles, and squares. The most important development in this area is DiffVG (Li et al., 2020), which removed the need for approximations and introduced techniques to handle antialiasing. They further pioneered image-supervised SVG generative models by training a Variational Autoencoder (VAE) and a Generative Adversarial Network (GAN; Goodfellow et al., 2014) on MNIST (LeCun et al., 1998) and QuickDraw (Ha & Eck, 2017). These generative capabilities have subsequently been extended in Im2Vec (Reddy et al., 2021), which adopts a VAE including a recurrent neural network to generate vector graphics as sets of deformed and filled circular paths, which are differentiably composited and rasterized, allowing for back-propagation of a multi-resolution MSE-based pyramid loss. However, all of these models lack versatile conditioning (such as text) and focus on either image vectorization, i.e., the task of creating the closest vector representation of a raster prior, or vector graphics interpolation.

A different type of SVG generation enabled by DiffVG is painterly rendering (Ganin et al., 2018; Nakano, 2019), where an algorithm iteratively fits a given set of vector primitives to match an image, guided by a deep perceptual loss function. To achieve this goal, CLIPDraw (Frans et al., 2022) rasterized a set of randomly initialized SVG paths and encoded these with a pre-trained CLIP (Radford et al., 2021) image encoder, iteratively minimizing the cosine distance between such embeddings and the text description. Vector Fusion (Jain et al., 2023) and SVGDreamer (Xing et al., 2024) leveraged Score Distillation Sampling (SDS; Poole et al., 2022) to induce abstract semantic knowledge from an off-the-shelf Stable Diffusion model (Rombach et al., 2022). Finally, some other approaches based on neural implicit representations model vector graphics as continuous functions encoded in neural networks (Thamizharasan et al., 2024; Zhang et al., 2024; Polaczek et al., 2025).

### 2.2. Vector Quantization

VQ-VAE (Van Den Oord et al., 2017) is a well-known improved architecture for training Variational Autoencoders (Kingma & Welling, 2013; Rezende et al., 2014). Instead of focusing on representations with continuous features as in most prior work (Vincent et al., 2010; Denton et al., 2016; Hinton & Salakhutdinov, 2006; Chen et al., 2016), the encoder in a VQ-VAE emits discrete rather than

---

[1] https://github.com/potpov/VectorGrimoire

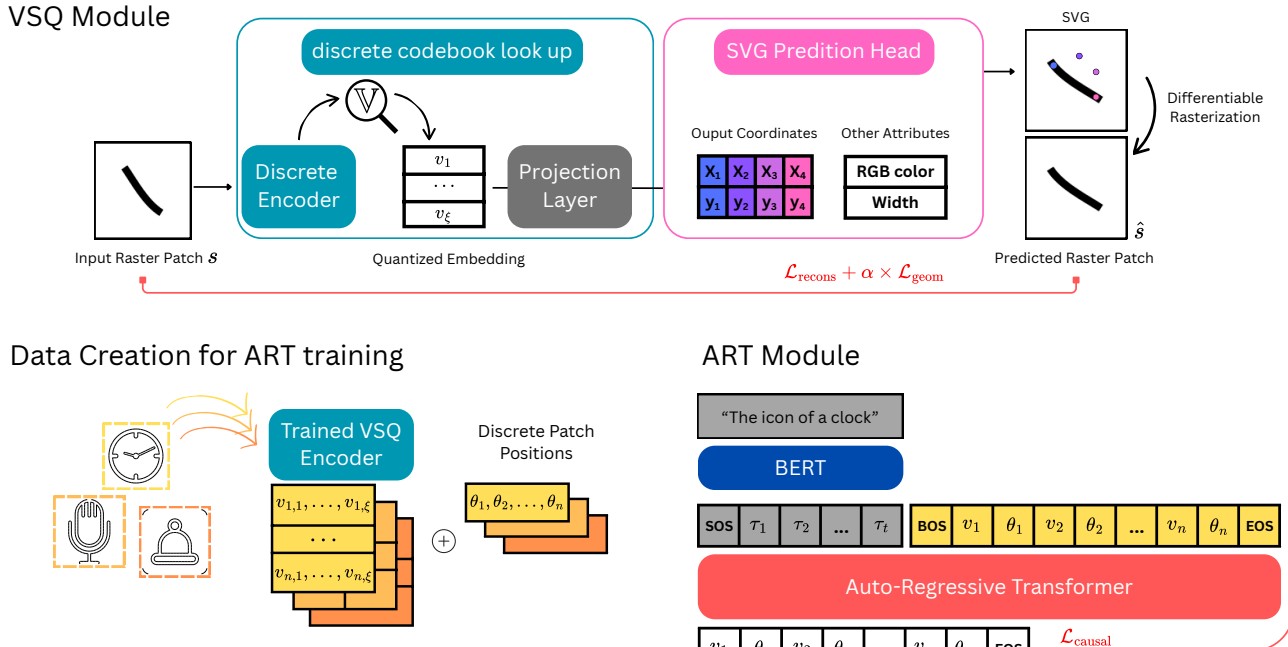

*Figure 2.* Overview of training pipeline of GRIMOIRE. First, the VSQ module encodes raster input patches into discrete codes and learns to reconstruct each patch as an SVG shape using visual supervision and optionally, a geometric constraint. After this, each image is encoded into a series of discrete codes using the trained VSQ encoder. Finally, the ART module learns the joint distribution of these codes and the corresponding text description. At inference, only text or partial codes are provided for conditioning the ART module.

continuous codes. Each code maps to the closest embedding in a codebook of limited size. The decoder learns to reconstruct the original input image from the chosen codebook embedding. Both the encoder–decoder architecture and the codebook are trained jointly. After training, the autoregressive distribution over the latent codes is learnt by a second model, which then allows for generating new images via ancestral sampling. Latent discrete representations were already pioneered in previous work (Mnih & Gregor, 2014; Courville et al., 2011). Mentzer et al. (2023) simplified the design of the vector quantization in VQ-VAE with a scheme called finite scalar quantization (FSQ), where the encoded representation of an image is projected to the nearest position on a low-dimensional hypercube. In this case, no additional codebook must be learned, but rather it is given implicitly, which simplifies the loss formulation. Our work builds in part on the VQ-VAE framework and includes the FSQ mechanism.

## 3. Method

### 3.1. Stage 1 – Visual Shape Quantizer

The first stage of our model employs a **V**isual **S**hape **Q**uantizer (VSQ), a vector-quantized auto-encoder, whose encoder $E_{\text{VSQ}}$ maps an input image $I$ onto a discrete codebook $\mathbb{V}$ through vector-quantization and decodes that quan-

tized vector into shape parameters of cubic Bézier curves through the decoder $D_{\text{VSQ}}$. Instead of learning the codebook (Van Den Oord et al., 2017), we adopt the more efficient approach of defining our codebook $V$ as a set of equidistant points in a hypercube with $q$ dimensions. Each dimension has $l$ unique values: $L = [l_1, l_2, \ldots, l_q]$. The size of the codebook $|\mathbb{V}|$ is hence defined by the product of values of all $q$ dimensions. We define $q = 5$ and $L = [7, 5, 5, 5, 5]$ for a target codebook size of 4,375 unique codes, following the recommendations of the original authors (Mentzer et al., 2023).

Before being fed to the encoder $E_{\text{VSQ}}$, each image $I \in \mathbb{R}^{C \times H \times W}$ is divided into patches $\mathbf{S} = (s_1, s_2, \ldots, s_n)$, with $s_i \in \mathbb{R}^{C \times 128 \times 128}$, where $C = 3$ is the number of channels. A set of discrete anchor coordinates $\mathbf{\Theta} = (\theta_1, \theta_2, \ldots, \theta_n)$ with $\theta_i \in \mathbb{N}^2$ being the center coordinate of $s_i$ in the original image $I$ is also saved. The original image $I$ can then be reconstructed using $S$ and $\Theta$.

We experiment on four datasets (see Section A.1). For MNIST, the patches are obtained by tiling each image in a $6 \times 6$ grid. For Fonts and FIGR-8, each patch depicts a part of the target outline. We also include preliminary results on Emojis, where our training data is created using the Segment Anything (SAM) model (Kirillov et al., 2023), which provides a series of masks for the entire image. In this extraction pipeline, each mask produces one patch. The

**Raw Images** **Extraction Method** **Extracted Patch and Positions**

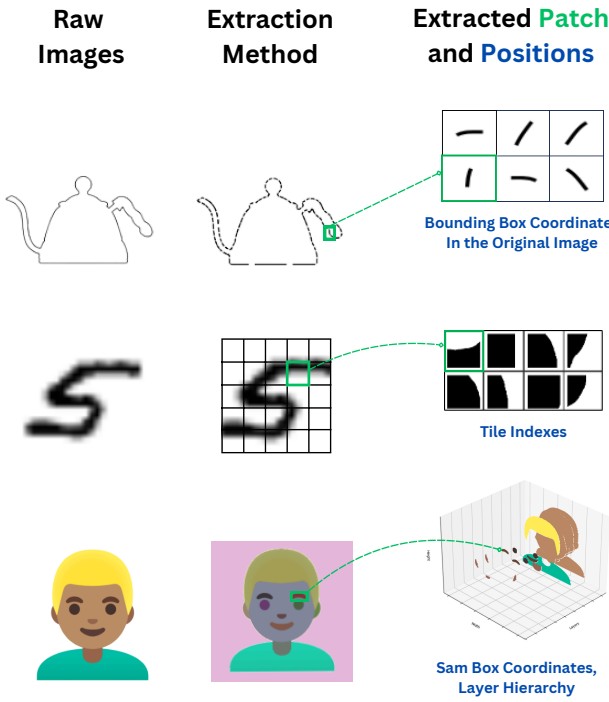

**Bounding Box Coordinates In the Original Image**

**Tile Indexes**

**Sam Box Coordinates, Layer Hierarchy**

*Figure 3.* Overview of the data generation process for GRIMOIRE. For FIGR-8, we extract the outlines of each icon and create small centered raster segments. We save the original anchor position of each segment for the second stage of our training pipeline. For the MNIST digits, we simply create patches from a $6 \times 6$ Grid and save the index of each tile. For Emoji, we generate an overall segmentation mask with SAM and create a raster patch from each mask. The mask bounding box coordinates can be used in the second stage similarly to the experiments on FIGR-8. The layer order is created by sorting patches by area. Fonts comes in vector format and can be easily manipulated to extract strokes, similarly to FIGR-8. More information about the outline extraction is provided in Section A.3.

different extraction approaches and the resulting patches and coordinates are depicted in Figure 3.

The VSQ encoder $E_{\text{VSQ}}$ maps each patch $s_i \in \mathbb{R}^{C \times 128 \times 128}$ to $\xi$ codes on the hypercube $E_{\text{VSQ}} : \mathbb{R}^{C \times 128 \times 128} \mapsto V$ as follows. Each centered raster patch $s_i$ is encoded with a ResNet-18 (He et al., 2016) into a latent variable $z_i \in \mathcal{Z} \subset \mathbb{R}^{d \times \xi}$ with $d = 512$. Eventually, each of the $\xi$ codes is projected to $q$ dimensions through a linear mapping layer and finally quantized, resulting in $\xi$ codes $(v_1, v_2, \ldots, v_\xi)$ with $v_i \in \mathbb{V}$.

The decoder $D_{\text{VSQ}}$ consists of a projection layer, which transforms all the $\xi$ predicted codes back into the latent space $\mathcal{Z}$, and a lightweight neural network $\Phi_{\text{points}}$, which predicts the control points of $\nu$ cubic Bézier curves that form a single connected path.

Finally, the predicted path of $\nu$ Bézier curves from $\Phi_{\text{points}}$ passes through the differentiable rasterizer to obtain a raster output $\hat{s}_i = \text{DiffVG}(D_{\text{VSQ}}(E_{\text{VSQ}}(s_i)))$. In order to learn to reconstruct strokes and shapes, we train the VSQ module using the mean squared error:

$$\mathcal{L}_{\text{recons}} = (s - \hat{s})^2. \tag{1}$$

$D_{\text{VSQ}}$ can be extended to predict continuous values for any visual attribute supported by the differentiable rasterizer. Hence, we also propose series of other fully-connected prediction heads that can optionally be enabled: $\Phi_{\text{width}} : \mathcal{Z} \mapsto \mathbb{R}$ predicts the stroke width of the overall shape, and $\Phi_{\text{color}} : \mathcal{Z} \mapsto \mathbb{R}^{\mathbb{C}}$ outputs the stroke color or the filling color for the output of $\Phi_{\text{points}}$. All the modules are followed by a sigmoid activation function.

While $\mathcal{L}_{\text{recons}}$ would suffice for training the VSQ, operating only on the visual domain could lead to degenerate strokes and undesirable local minima. To mitigate this, we propose a novel geometric constraint $\mathcal{L}_{\text{geom}}$, which punishes the placement of control point at irregular distances measured between all combinations of points predicted by $\Phi_{\text{points}}$.

Let $P = (p_1, p_2, ..., p_{\nu+1})$ be the set of all start and end points of a curve with $p_i = (p_i^x, p_i^y)$ and $p_i^x, p_i^y \in [0, 1]$. Then $\rho_{i,j}$ is defined as the Euclidean distance between two points $p_i$ and $p_j$, $\overline{\rho}_j$ is defined as the mean scaled inner distance for point $p_j$ to all other points in $P$, while $\delta_j$ is the average squared deviation from that mean for point $p_j$:

$$\overline{\rho}_j = \frac{1}{\nu} \sum_{\substack{i=1 \\ i \neq j}}^{\nu+1} \frac{\rho_{i,j}}{|i-j|} \qquad \delta_j = \frac{1}{\nu} \sum_{\substack{i=1 \\ i \neq j}}^{\nu+1} \left( \frac{\rho_{i,j}}{|i-j|} - \overline{\rho}_j \right)^2 \tag{2}$$

$\mathcal{L}_{\text{geom}}$ is finally defined as the average of the deviations for all start and end points in $P$. $\mathcal{L}_{\text{geom}}$ is then weighted with $\alpha$ and added to the reconstruction loss.

$$\mathcal{L}_{\text{geom}} = \frac{1}{\nu+1} \sum_{j=1}^{\nu+1} \delta_j \qquad \mathcal{L}_{\text{VSQ}} = \mathcal{L}_{\text{recons}} + \alpha \times \mathcal{L}_{\text{geom}} \tag{3}$$

Here, $\alpha$ is a hyperparameter. The overall scheme of GRI-MOIRE including the first stage of training is depicted in Figure 2.

### 3.2. Stage 2 – Auto-Regressive Transformer

After the VSQ is trained, each patch $s_i$ can be mapped onto a code $v_i$ of the codebook $V$ using the encoder $E_{\text{VSQ}}$ and the quantization method. However, the predicted patch $\hat{s}_i$ captured by the VSQ does not describe a complete SVG, as the centering leads to a loss of information about their

global position $\theta_i$ on the original canvas. Also, the sequence of tokens is still missing the text conditioning. This is addressed in the second stage of GRIMOIRE. The second stage consists of an **A**uto-**R**egressive **T**ransformer (ART) that learns for each image $I$ the joint distribution over the text, positions, and stroke tokens. A textual description $T$ of $I$ is tokenized into $\mathcal{T} = (\tau_1, \tau_2, \ldots, \tau_t)$ using a pre-trained BERT encoder (Devlin et al., 2018) and embedded. $I$ is visually encoded by transforming its patches $s_i$ onto $v_i \in V$ via the encoder $E_{\text{VSQ}}$, whereas each original patch position $\theta_i \in \Theta$ is mapped into the closest position in a $256 \times 256$ grid, resulting in $256^2$ possible position tokens. Special tokens <SOS>, <BOS>, and <EOS> indicate the start of a full sequence, beginning of the patch token sequence, and end of sequence, respectively. Patch tokens alternate with corresponding position tokens. The final input sequence for a given image to the ART module becomes:

$$x = (\texttt{<SOS>}, \tau_1, \ldots, \tau_t, \texttt{<BOS>}, \theta_1, v_1, \ldots \theta_n, v_n, \texttt{<EOS>})$$

The total amount of representable token values then has a magnitude of $|V| + 256^2 + 3 = 69,914$ for $|V| = 4{,}375$. A learnable weight matrix $W \in \mathbb{R}^{d \times 69{,}914}$ embeds the position and visual tokens into a vector of size $d$. The BERT text embeddings are projected into the same $d$-dimensional space using a trainable linear mapping layer. The ART module consists of 12 and 16 standard Transformer decoder blocks (with causal multi-head attention using 8 attention heads) for fonts and icons, respectively. The final loss for the ART module is defined as:

$$\mathcal{L}_{\text{Causal}} = -\sum_{i=1}^{N} \log p(x_i \mid x_{<i}; \theta) \qquad (4)$$

During inference, the input to the ART module is represented as $x = (\texttt{<SOS>}, \tau_1, \ldots, \tau_t, \texttt{<BOS>})$, where new tokens are predicted auto-regressively until the <EOS> token is generated. Additionally, visual strokes can be incorporated into the input sequence to condition the generation process.

## 4. Experimental Setting

When training on FIGR-8, we utilize a contour-finding algorithm (Lorensen & Cline, 1987) to extract outlines from raster images, which are then divided into several shorter segments. Additional details regarding this extraction process can be found in Section A.3. In contrast, the Fonts dataset is natively available in vector format, making it easier to manipulate, similar to icons, before undergoing rasterization.

We propose two variants of $\Phi_{\text{points}}$ described in Section 3.1, a fully-connected neural network $\Phi_{\text{points}}^{\text{stroke}} : \mathcal{Z} \mapsto$

$\mathbb{R}^{(2 \times (\nu \times 3 + 1))}$, which predicts connected strokes, and a 1-D CNN $\Phi_{\text{points}}^{\text{shape}} : \mathcal{Z} \mapsto \mathbb{R}^{(2 \times (\nu \times 3))}$, which outputs a closed shape.

We use $\mathcal{L}_{\text{geom}}$ only for the experiments with $\Phi_{\text{points}}^{\text{stroke}}$ and set $\alpha = 0.4$. We opt to train the ResNet encoder from scratch during this stage, since the target images belong to a very specific domain. The amount of trainable parameters is $15.36M$ for the encoder and $0.8M$ for the decoder. The final inference pipeline discards the encoder and only requires the lightweight trained decoder $D_{\text{VSQ}}$, hence resulting in faster inference.

## 5. Results

In this section, we report the results for all our experiments and discuss our findings. First, in Section 5.1 and Section 5.2, we examine the **quality** of the reconstructions and generations produced by GRIMOIRE in comparison to existing methods. Then, in Section 5.3 we highlight the **flexibility** of our approach, demonstrating how GRIMOIRE can be easily extended to incorporate additional SVG features and more complex targets. Furthermore, in Section 5.4 we compare GRIMOIRE with some popular SDS-based method

### 5.1. Reconstructions

**Closed Paths**. We begin by presenting the reconstruction results of our VSQ module on the MNIST dataset. In our experiments, we model each patch shape using a total of 15 segments. Increasing the number of segments beyond this point did not yield any significant improvement in reconstruction quality. Given the simplicity of the target shapes, we adopted a single code per shape.

We also conducted a comparative analysis of the reconstruction capabilities of our VSQ module against Im2Vec. To assess the generative quality of our samples, we employed the Fréchet Inception Distance (FID; Heusel et al., 2017) and CLIPScore (Radford et al., 2021), both of which are computed using the image features of a pre-trained CLIP encoder. Additionally, to validate our VSQ module, we considered the reconstruction loss $\mathcal{L}_{\text{recons}}$, as it directly reflects the maximum achievable performance of the network and provides a more reliable metric.

As shown in Table 1, our VSQ module consistently achieves lower reconstruction errors than Im2Vec across all MNIST digits. In Table 2, we also evaluate a subset with only digits *zero*, chosen for its challenging topology, where our method again outperforms Im2Vec. For MNIST, we fill Im2Vec's predicted shapes to match the raster ground truth, while for other scenarios, we report both filled and unfilled versions.

Our reconstructions achieve higher CLIPScores in all cases. The only exception is FID, where Im2Vec occasionally per-

*Table 1.* Results for reconstructions of GRIMOIRE and Im2Vec on the test-set including all classes. The last row includes post-processing.

| | MNIST | | | Fonts | | | FIGR-8 | | |
|---|---|---|---|---|---|---|---|---|---|
| Model | MSE ⇊ | FID ⇊ | CLIP ⇈ | MSE ⇊ | FID ⇊ | CLIP ⇈ | MSE ⇊ | FID ⇊ | CLIP ⇈ |
| Im2Vec (filled) | 0.140 | **1.33** | 25.02 | 0.140 | 2.04 | 26.82 | 0.330 | 16.10 | 26.17 |
| Im2Vec | *n/a* | *n/a* | *n/a* | 0.050 | 5.64 | 26.72 | 0.050 | 13.90 | 26.17 |
| VSQ | **0.090** | 7.09 | **25.24** | 0.014 | 4.45 | 28.61 | 0.004 | 1.42 | 31.09 |
| VSQ + PI | *n/a* | *n/a* | *n/a* | **0.011** | **0.29** | **28.96** | **0.002** | **0.05** | **32.03** |

*Table 2.* Results for reconstructions of GRIMOIRE and Im2Vec on the test-set, using the class reported next to the dataset name. The last row includes post-processing.

| | MNIST (0) | | | Fonts (A) | | | Icons (Star) | | |
|---|---|---|---|---|---|---|---|---|---|
| Model | MSE ⇊ | FID ⇊ | CLIP ⇈ | MSE ⇊ | FID ⇊ | CLIP ⇈ | MSE ⇊ | FID ⇊ | CLIP ⇈ |
| Im2Vec (filled) | 0.218 | **2.20** | 24.61 | 0.087 | 1.64 | 26.27 | 0.120 | 2.40 | 30.90 |
| Im2Vec | *n/a* | *n/a* | *n/a* | 0.060 | 6.33 | 25.78 | 0.110 | 11.17 | 30.40 |
| VSQ | **0.130** | 11.2 | **26.68** | 0.020 | 4.50 | 29.13 | 0.002 | 1.26 | 31.64 |
| VSQ + PI | *n/a* | *n/a* | *n/a* | **0.012** | **0.61** | **29.46** | **0.001** | **0.07** | **32.94** |

forms better. We attribute this to the lower resolution of the ground truth images, which affects FID stability. CLIPScore mitigates this by directly measuring similarity to the textual description.

**Strokes**. We analyze VSQ reconstruction errors on Fonts and FIGR-8 under varying segment counts, codes per shape, and input stroke lengths. For Fonts, using more than one segment per shape consistently degrades reconstruction, likely because the dataset's strokes require fewer Bézier curves. Shorter stroke length thresholds improve reconstruction, with MSE decreasing as the threshold moves from $11\%$ to $7\%$ to $4\%$ of the image size. While shorter strokes are easier to model, overly short settings may cause worse predictions.

The best reconstructions are achieved by using multiple codes per centered stroke. The two-codes configuration has an average decrease in MSE of 18.28%, 41.46%, and 26.09% for the respective stroke lengths. However, the best-performing configuration with two codes per shape is just 11.36% better than the best single code representative, which we believe does not justify twice the number of required visual tokens for the second stage training. Throughout our experiments, the configurations with multiple segments consistently benefit from our geometric constraint. Ultimately, for our final experiments, we choose ($\nu = 2$, $\xi = 1$) for Fontsand ($\nu = 4$, $\xi = 2$) for FIGR-8.

Regarding the comparison with Im2Vec, Table 2 shows that the text-conditioned GRIMOIRE on a single glyph or icon has superior reconstruction performance even if Im2Vec

is specifically trained on such subset. In Table 1, we also report the values after training on the full datasets. In this case, GRIMOIRE substantially outperforms Im2Vec, which is unable to cope with the complexity of the data.

Finally, as GRIMOIRE quickly learns to map basic strokes or shapes onto its finite codebook and due to the similarities between those primitive traits among various samples in the dataset, we find GRIMOIRE to converge even before completing a full epoch on any dataset. Despite the reconstruction error being considerably higher, we also notice reasonable domain transfer capabilities between FIGR-8 images and Fonts when training the VSQ module only on one dataset and keeping the maximum stroke length consistent. Qualitative examples of the re-usability of the VSQ module are reported in the Appendix.

### 5.2. Generations

**Text Conditioning.** We compare GRIMOIRE with Im2Vec by generating glyphs and icons and handwritten digits, and report the results in Table 3. Despite Im2Vec being tailored for single classes only, our general model shows superior performance in CLIPScore for all datasets. Im2Vec shows a generally lower FID score in the experiments with filled shapes, which we attribute again to the lower resolution of the ground truth images (MNIST) and a bias in the metric itself as CLIP struggles to produces meaningful visual embeddings for sparse images (Chowdhury et al., 2022) as for Fontsand FIGR-8. In contrast, in the generative results on

unfilled shapes, GRIMOIRE almost consistently outperforms Im2Vec by a large margin for glyphs and icons.

Note that we establish new baseline results for the complete datasets, as Im2Vec does not support text or class conditioning.

Looking at qualitative samples in Figure 1, one can see that contrary to the claim that surplus shapes collapse to a point (Reddy et al., 2021), there are multiple redundant shapes present in the generations of Im2Vec. A single star might then be represented by ten overlapping almost identical paths.

Overall, GRIMOIRE produces much cleaner samples with less redundancy, which makes them easier to edit and more visually pleasing. The text conditioning also allows for more flexibility. Furthermore, the generations are diverse, as can be seen in Figure 4, where we showcase multiple generations for the same classes from FIGR-8. Additional generations on all datasets are provided in the Appendix.

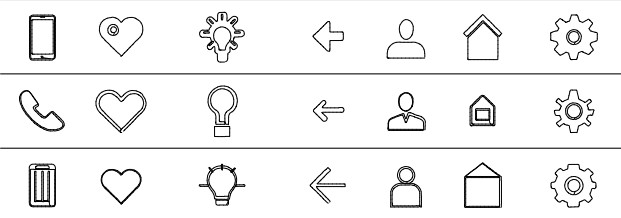

Phone  Heart  Light bulb  Arrow  User  Home  Settings

*Figure 4.* Examples of text-conditioned icon generation from GRI-MOIRE.

**Vector Conditioning.** We also evaluate GRIMOIRE on another task previously unavailable for image-supervised vector graphic generative models, which is text-guided icon completion. Figure 5 shows the capability of our model to complete an unseen icon, based on a set of given context strokes that start at random positions. GRIMOIRE can meaningfully complete various amounts of contexts, even when the strokes of the context stem from disconnected parts of the icon. We provide a quantitative analysis in Section A.13. The results in this section are all obtained with the default pipeline that post-processes the generation of our model. A detailed analysis of our post-processing is provided in Section A.4 and Section A.5.

### 5.3. Flexibility

We demonstrate the flexibility of GRIMOIRE with qualitative results on new SVG attributes. A key advantage of our two-stage generative pipeline is that the ART module remains independent of visual attributes, allowing the VSQ vector prediction head to be extended to any attribute supported by the differentiable rasterizer. Specifically, we enable stroke width and color prediction ($\Phi_{\text{width}}$ and $\Phi_{\text{color}}$), training the

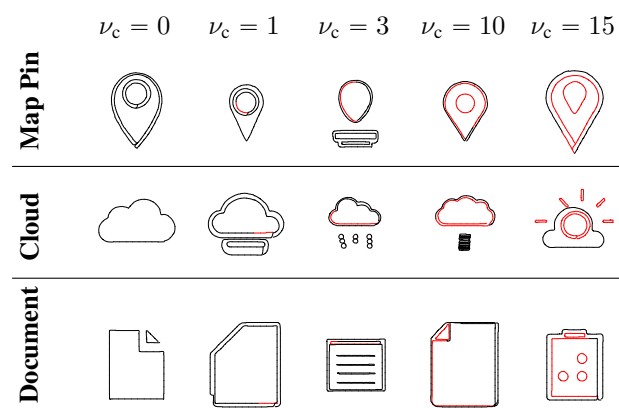

*Figure 5.* Different completions with varying number of context segments $\nu_c$ (marked in red). GRIMOIRE can meaningfully complete irregular starting positions of the context strokes.

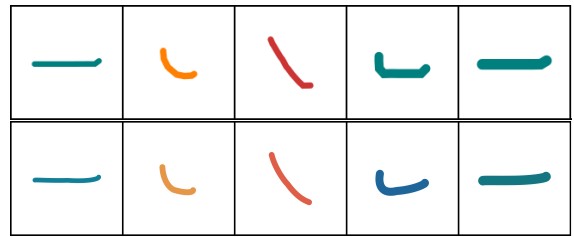

*Figure 6.* Inputs (top) and corresponding reconstructions (bottom) generated by a VSQ model trained to predict additional visual attributes of the input strokes, such as color and stroke width. Input from the test-set.

VSQ module on input patches with varying attributes. Figure 6 shows the results, where strokes are randomly colored using an eight-color palette and variable widths. The VSQ accurately learns these features without expanding the codebook or modifying the network architecture.

A similar evaluation on closed shapes (Figure 7) confirms that VSQ jointly encodes shape and color within a single code. In contrast, other vector generative models often rely on rigid tokenization schemes, making extension to new attributes more complex.

Finally, we report preliminary results of GRIMOIRE on more complex colorful targets based on a segmentation-based extraction approach. We validate this by reconstructing images of Emoji, following the MNIST setup but with a crucial difference: each closed shape represents an entire layer of the canvas rather than a tile. This allows for SVGs structured as editable layers, similarly to real-world use cases. Figure 8 illustrates per-layer reconstructions, while Figure 9 presents the final composited outputs. Further details on these experiments are reported in Section A.2.

*Table 3.* Results for generation using GRIMOIRE and Im2Vec. GRIMOIRE is trained with all the classes of the dataset and conditioned to the respective class using the text description. FID uses test-data as a target.

| Model | MNIST (0) | | MNIST (Full) | | Fonts (A) | | Fonts (Full) | | FIGR-8(Star) | | FIGR-8(Full) | |
|---|---|---|---|---|---|---|---|---|---|---|---|---|
| | FID ⇊ | CLIP ⇈ | FID ⇊ | CLIP ⇈ | FID ⇊ | CLIP ⇈ | FID ⇊ | CLIP ⇈ | FID ⇊ | CLIP ⇈ | FID ⇊ | CLIP ⇈ |
| Im2Vec (filled) | **2.22** | 24.69 | *n/a* | *n/a* | **1.20** | 25.81 | *n/a* | *n/a* | **2.97** | 31.72 | *n/a* | *n/a* |
| Im2Vec | *n/a* | 25.21 | *n/a* | *n/a* | 5.36 | 25.39 | *n/a* | *n/a* | 11.59 | 31.88 | *n/a* | *n/a* |
| GRIMOIRE (ours) | 12.25 | **26.60** | **9.25** | **25.25** | 5.61 | **30.60** | 1.67 | **28.64** | 6.25 | **32.24** | 0.64 | **29.00** |

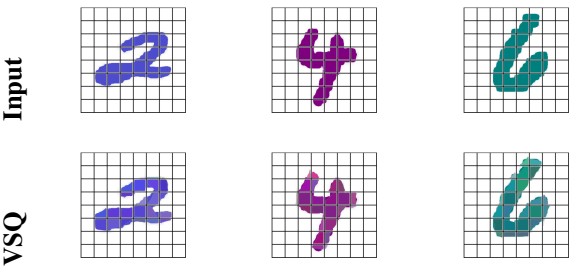

*Figure 7.* Reconstruction of MNIST digits when the VSQ module also predicts the filling color. Input from the test-set.

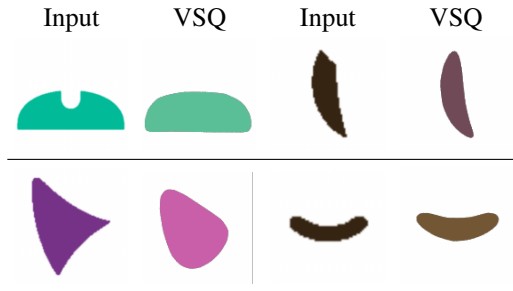

*Figure 8.* Reconstructions of individual layers of emojis from the our VSQ. Input from the test-set.

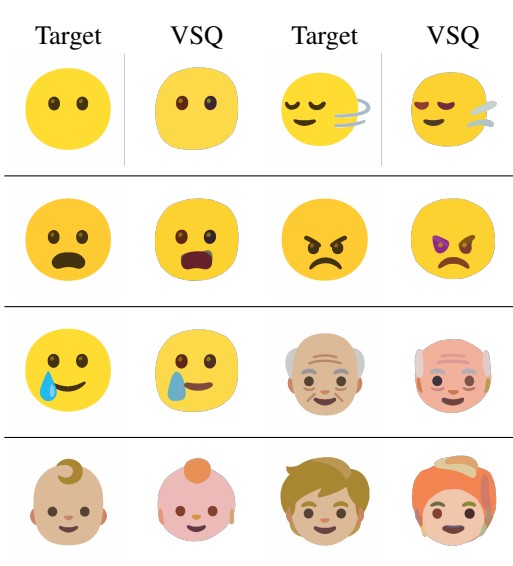

*Figure 9.* Reconstructions of emojis from the our VSQ, all the SVG layers are rendered together. Input from the test-set.

### 5.4. Differences with SDS-based Methods

In this section, we clarify how our model differs from popular Score Distillation Sampling (SDS) architectures.

**Lack of target.** SDS methods do not involve training, and rely on pretrained backbones, which produce more artistic and visually-appealing results, but also unbound to any specific target data. SDS methods lack any control on the target domain. To highlight this aspect, Figure 10 reports an example of class images adopted in this work, and shows the different generations obtained with GRIMOIRE and popular SDS methods such as VectorFusion, CLIPDraw, and SVGdreamer. GRIMOIRE produces simple yet diverse generations that are coherent with its reference dataset. In contrast, in all cases, the generations from SDS based methods appear distant from the target distribution, often partially ignoring the "black and white" suffix in the prompt.

Making this analysis quantitative is not straightforward. The FID score with regard to the image distribution is reliable on thousands of samples, but the computational cost of SDS-based models requires up to hours for a few samples (e.g., SVGDreamer) or uses costly proprietary models (e.g., Chat2SVG). We have hence used the PSNR on 20 generated samples from all models. The results in Table 4 highlight how all models fall short on our dataset distribution.

| Model | PSNR (dB) |
|---|---|
| GRIMOIRE (ART) | **45.19** |
| CLIPDraw | 28.68 |
| VectorFusion | 36.62 |
| SVGDreamer | 34.53 |

*Table 4.* Average PSNR of 20 generated samples for the class "User" with GRIMOIRE and existing SDS-based methods.

**Speed.** SDS methods are also iterative by design, which means that generating results is extremely slow in practice. One of the motivations for training a generative pipeline like GRIMOIRE is that, at inference time, producing a new sample merely takes the time of a forward pass. Indeed,

GRIMOIRE is two orders of magnitude faster than popular SDS based methods. In Table 5, we report the generation time (time for inference and file saving) for an image with Grimoire, VectorFusion, and CLIPdraw. These values were obtained across 20 generations on one NVIDIA H100.

| Model | Generation Time (s) |
|---|---|
| GRIMOIRE (ART) | **2.34** |
| CLIPDraw | 100.19 |
| VectorFusion | 379.74 |
| SVGDreamer | 250.22 |

*Table 5.* Average generation times of an icons given only the text prompt measured on five samples using one NVIDIA H100 GPU. SDS-based methods are extremely slow due to their iterative optimization strategy and result impractical for real-life applications.

*Figure 10.* Text-conditioned generations from GRIMOIRE and SDS-based methods for the "Heart" class. The first row shows samples from FIGR-8. For CLIPDraw, VectorFusion and SVGDreamer, we used the prompt: "The icon of a heart, black and white.

## 6. Conclusion

This work introduces GRIMOIRE, a novel framework for generating and completing complex SVGs using only raster image supervision. GRIMOIRE enhances the output quality over existing raster-supervised SVG models, while enabling flexible, text-conditioned generation. We validate it on filled shapes using a tile-patching or segmentation-based strategy and on strokes with fonts and icons datasets. Our results show superior performance compared to existing models, even when adapted to specific image classes. We show that GRIMOIRE can be seamlessly extended to support new SVG attributes when included in the training data.

## Acknowledgments

This work was conducted in part with funding from the HPI–MIT Designing for Sustainability Program.

## Impact Statement

This work aims to advance the field of text-to-SVG generation by introducing a new framework that learns using only raster image supervision. GRIMOIRE enables broader accessibility to training data. While the method may benefit applications in education, design, and accessibility, we do not foresee any immediate ethical concerns or adverse societal impacts stemming from this research. Potential indirect effects are similar as for other Generative AI techniques. We hope that models that emit vector graphics can make it easier for graphic designers to build upon, enabling improved human–AI workflows.

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

# A. Appendix

The appendix is organized as follows. In Section A.1, we provide information about the data used in our work; in Section A.2, we report additional details on the segmentation-based extraction method; in Section A.3 and Section A.4, we describe the pre-processing and post-processing applied in our experiments; in Section A.5, we show results with different post-processing methods; in Section A.6, we provide a comparison with vector-supervised methods; in Section A.7, Section A.8, and Section A.9, we study sensitivity with regard to patch size, stroke length, and codebook size, respectively; in Section A.10, we report results of Im2Vec on other classes; in Section A.11, we show additional qualitative results on the effects of our geometric loss; in Section A.12, we share some implementation details; in Section A.13, we show more results on GRIMOIRE conditioned strokes; in Section A.14, we explore domain transfer capabilities; in Section A.15, we analyze the codebook usage of our model; finally in Section A.17 and Section A.18, we report more qualitative results and comparison on the reconstruction and generative of GRIMOIRE. We also report a glossary of all the notation in Section A.19.

## A.1. Dataset

**MNIST.** We conduct our initial experiments on the MNIST dataset (LeCun et al., 1998). We upscale each digit to $128 \times 128$ pixels and generate the textual description using the prompt "*x* in black color", where *x* is the class of each digit. We adopt the original train and test splits.

**Fonts.** For our experiments on fonts, we use a subset of the SVG-Fonts dataset (Lopes et al., 2019). We remove fonts where capital and lowercase glyphs are identical, and consider only 0–9, a–z, and A–Z glyphs, which leads to 32,961 unique fonts for a corpus of ∼2M samples. The font features – such as type of character or style – are extracted from the *.TTF* file metadata. The final textual description for a sample glyph $g$ in font style $s$ is built using the prompt: "[capital] $g$ in $s$ font", where "capital " is included only for the glyphs A-Z. We use 80%, 10%, and 10% for training, testing, and validation respectively.

**FIGR-8.** We validate our method on more complex data and further use a subset of FIGR-8 (Clouâtre & Demers, 2019), where we select the 75 majority classes (excluding "arrow") and any class that contains those, e.g., the selection of "house" further entails the inclusion of "dog house". This procedure yields 427K samples, of which we select 90% for training, 5% for validation, and 5% for testing. We use the class names as textual descriptions without further processing besides minor spelling correction. Since the black strokes of FIGR-8 mark the background rather than the actual icon, we invert the full dataset before applying our additional pre-processing described in Section A.3.

**Emoji.** For our preliminary experiments with segmentation-guided patch extraction, we use a subset of standard emoji images (emoji dataset, 2022). Specifically, we focus on images that primarily depict faces, selecting 107 for training and 20 for the test.

## A.2. Segmentation-Guided Patch Extraction

In this section, we provide more information and example results on emoji generation. The model setup is similar to the experiments presented for the MNIST dataset with one fundamental difference: Each predicted closed shape targets one layer of the entire image canvas instead of a tile. This setting enables the prediction of a final SVG that resembles real-world use cases where vector data is a set of editable layers, ultimately composited altogether. Our training data is created using the Segment Anything (SAM) model from Meta, which provides a series of masks for the entire image. In our extraction pipeline, each mask produces one layer. We quantize the original image into 4,096 possible colors and create a raster layer for each mask by using the median color in the original image for its respective mask. A qualitative example of the results from the extraction pipeline is shown in Figure 11. The image also depicts a three-dimensional visualization of the final extracted layers sorted by their area.

Each layer is center-cropped based on the bounding boxes of the SAM mask. White padding of 10 pixels is added on all sides similarly to what was done for the MNIST. However, in this scenario, padding does not create any artifact and merely becomes an additional scaling factor, since the reconstructed shapes fit the whole image size. During VSQ training, the ground truth cropping bounding boxes are used to scale and shift back the points predicted by the VSQ into the original position. These shifting values and the hierarchy of the layers become the new target of the ART module. We introduced minor additional changes to cope with the increasing complexity of the data, especially the color imbalance due to the small number of samples: The VSQ module outputs RGB colors per shape, but the raster and ground truth layers are converted into the CIE-LAB color space before computing the loss. The color channels (AB) of each layer are weighted inversely to

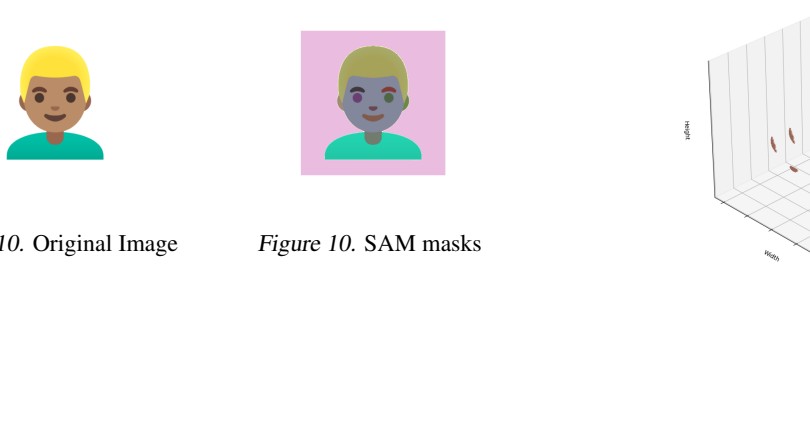

Figure 10. Original Image       Figure 10. SAM masks

Figure 10. Original Image       Figure 10. SAM masks

Figure 11. Layer extraction with SAM.

the frequency of the target color in the dataset. No weights are needed for the luminance channel. Figure 12 reports some examples of VSQ reconstructions by layer.

Notably, this reconstruction was achieved after training on only 110 emojis, and the results come from the test set. Common shapes (such as circles) and colors (such as yellow) are quickly learned, whereas more complicated shapes remain challenging (e.g., shapes of the hair). Overall, this is already a large improvement to other raster-supervised SVG generative models. Im2Vec does not learn the colors. As stated in the original Im2Vec paper and found in the repository, it uses colors hard-coded to reflect the target image (e.g., one yellow and three black shapes when the target is a simple emoji).

### A.3. Pre-Processing

This section provides additional information regarding the pre-processing and extraction techniques on the employed datasets.

**Shapes.** No pre-processing is conducted for the MNIST dataset. Images are simply tiled using a $6 \times 6$ grid and the central position of each tile in the original image is saved. For the Emoji dataset, each image is first processed with the SAM model to generate masks covering every pixel. We filter out masks with very small areas or multiple connected components. Finally, we convert the remaining masks into layer patches, assigning the foreground the median pixel value from the original image.

**Strokes.** For the FIGR-8 dataset, the pixels outlining the icons are isolated using a contour finding algorithm (Lorensen & Cline, 1987) and the coordinates are then used to convert them into vector paths. This simple procedure available in our code repository allows us to efficiently apply a standard pre-processing pipeline defined in Carlier et al. (2020) and already adopted by other studies (Wu et al., 2023; Tang et al., 2024). The process involves normalizing all strokes and breaking them into shorter units if their length exceeds a certain maximum percentage of the image size. Finally, each resulting path

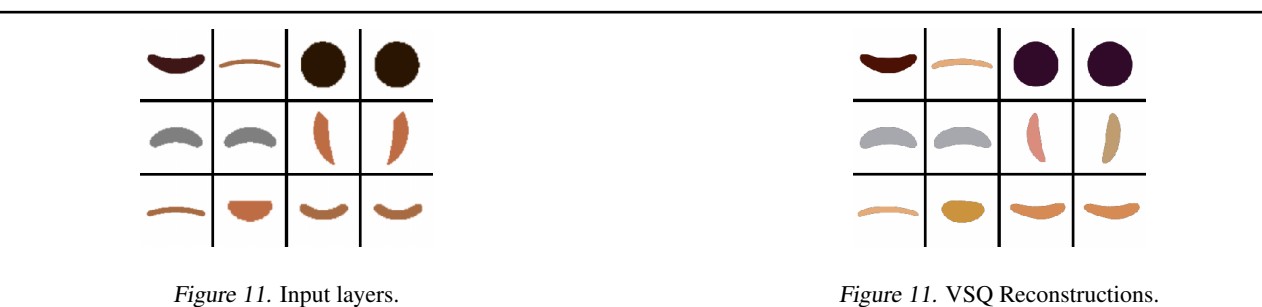

*Figure 11.* Input layers.            *Figure 11.* VSQ Reconstructions.

*Figure 12.* Inputs (left) input layers for the VSQ, (right) reconstructions of the model.

fragment is scaled, translated to the center of a new canvas $s$ by placing the center of its bounding box onto the center of $s$, and rasterized to become part of the training data. Since strokes in $S$ are all translated around the image center, the original center position $\theta$ of the bounding box in $I$ is recorded for each $s$ and saved. These coordinates are discretized in a range of $256 \times 256$ values. This approach is also used for Fonts, but since the data comes in vector format, there is no need for contour finding.

### A.4. Post-Processing

Our approach introduces small discrepancies with the ground truth data during tokenization. The VSQ introduces small inaccuracies in the reconstruction of the stroke, and the discretization of the global center positions may slightly displace said strokes. The latter serve as the training data for the auto-regressive Transformer and therefore represent an upper limit to the final generation quality. Similarly for MNIST, the use of white padding on each patch to facilitate faster convergence results in small background gaps when rendering all shapes together. These small errors compound for the full final image and may become fairly visible in the reconstructions.



*Figure 13.* Different SVG post-processing methods visualized. From left to right: raw generation, results of applying PC and PI, results of applying PC and PI by only considering nearest neighbors of consecutive strokes.

While we opted not to modify the global reconstructions of MNIST generation, for FIGR-8 and Fonts, we make use of SVG post-processing similar to prior work (Tang et al., 2024), which introduced Path Clipping (PC) and Path Interpolations (PI). In PC, the beginning of a stroke is set to the position of the end of the previous stroke. In PI, a new stroke is added that connects them instead. As we operate on visual supervision, the ordering of the start and end point of a stroke is not consistent. Hence, we adapt these two methods to not consider the start and end point, but rather consider the nearest neighbors of consecutive strokes. We also add a maximum distance parameter to the post-processing in order to avoid intentionally disconnected strokes to get connected. See Figure 13, Figure 14 for a qualitative depiction of this process and Section A.5 for a quantitative comparison.

### A.5. Results with Different Post-Processing

In GRIMOIRE, the resulting full vector graphic generation is characterized by fragmented segments. This is because the output strokes of the VSQ decoder are each locally centered onto a separate canvas, and the auto-regressive Transformer, which is responsible for the absolute position of each shape, returns only the center coordinates of the predicted shape without controlling the state of connection between different strokes. To cope with this, in Section A.4, we introduced several post-processing algorithms. In this section, we report additional information about the performance of each of

them for the VSQ module (reconstruction) and the overall GRIMOIRE (generation). Table 6 shows that the PC technique consistently outperforms the alternatives across both datasets in terms of both FID and CLIPScore.

| Model | Fonts | | | FIGR-8 | | |
|---|---|---|---|---|---|---|
| | MSE | FID | CLIP | MSE | FID | CLIP |
| VSQ | 0.0144 | 4.45 | 28.61 | 0.0045 | 1.29 | 31.17 |
| VSQ (+PC) | 0.0135 | **0.23** | **29.24** | **0.0023** | 0.10 | 31.97 |
| VSQ (+PI) | **0.0106** | 0.29 | 28.96 | 0.0028 | **0.07** | **32.0** |
| GRIMOIRE | *n/a* | 4.44 | 28.45 | *n/a* | 4.20 | 26.96 |
| GRIMOIRE (+PC) | *n/a* | **1.67** | **28.64** | *n/a* | **3.58** | **27.45** |
| GRIMOIRE (+PI) | *n/a* | 1.86 | 28.43 | *n/a* | 4.57 | 26.73 |

*Table 6.* Reconstruction capabilities of our VSQ module and generative performance of GRIMOIRE with different post-processing techniques after training on Fonts and FIGR-8.

## A.6. Comparison with Vector-supervised Methods

We extended our analysis to two vector-supervised methods – DeepSVG and IconShop – training them on the same FIGR-8 data used for GRIMOIRE. Unlike GRIMOIRE, DeepSVG supports conditioning only on class identifiers; therefore, we assigned a unique identifier to each class in FIGR-8.

We also finetuned Llama 3.2 on FIGR-8 with minimal data pre-processing. We believe this to be an insightful analysis that shows how tailored tokenization pipelines and extensive data pre-processing are necessary for other vector-supervised models to perform effectively. We wish to highlight how raster and vector data provide very different supervising signals.

**Llama**. We fine-tuned Llama (instruction tuning) for three days on eight H100 GPUs. Minimal pre-processing includes rounding up the path coordinates to integer values. Upon inspection, this did not affect the quality of the image. We use the original chat template and included special tokens to delimit the SVG code. The performance at inference appears very poor. The model predicts the most recurrent patterns in the dataset, resulting mainly in circular artifacts. The SVG syntax is, however, correct most of the time, allowing rendering.

**DeepSVG**. We train DeepSVG using the official training script. The model converges within a few hours, but the results are also not good, yielding the lowest CLIPScore and FID among all models.

**IconShop**. We also re-trained the original IconShop model on the subset of FIGR-8 used in Grimoire. In this case, the performance of the model is comparable to Grimoire, resulting in slightly better CLIPscore and FID.

All results are reported in Table 7.

| **Model** | **CLIPScore** | **FID** | **Conditioning** | **Supervision** |
|---|---|---|---|---|
| DeepSVG | 22.10 | 58.03 | Class | Vector |
| Llama 3.2 | 25.45 | 38.93 | Prompt | Vector |
| Grimoire | 29.00 | 0.64 | Prompt | Raster |
| IconShop | 31.18 | 0.40 | Prompt | Vector |

*Table 7.* Comparison of GRIMOIRE with vector-supervised methods. Llama was trained with almost minimal pre-processing.

## A.7. Analysis of Patch and Grid Sizes

In this section, we report results after training the VSQ module on MNIST using varying grid sizes (default: 5) and patch sizes (default: $128 \times 128$). The patch size had a limited effect on performance, while increasing the number of tiles (i.e., smaller patches) improved results, likely due to the simpler topology. In all configurations, the reconstruction error remained below that of Im2Vec. Table 8 reports MSE on the test set.

| Patch Size | Tiles = 3 | Tiles = 5 | Tiles = 8 |
|---|---|---|---|
| 32 | 0.093 | 0.092 | 0.078 |
| 64 | 0.092 | 0.090 | **0.071** |
| 128 | 0.090 | 0.094 | 0.078 |

*Table 8.* Effect of patch and grid size on performance.

## A.8. Analysis of Stroke Length

To assess the impact of stroke properties on VSQ performance, we conducted two additional experiments:

**Stroke length variations**: We created patches with smaller or larger strokes. Results show that shorter strokes yield lower reconstruction errors, similarly to the grid size variations.

**Multiple stroke predictions per patch**: We extended the prediction head of the VSQ to output two strokes per patch instead of one (as in the paper). The results show that more than one segment per shape consistently degrades the reconstruction quality. This suggests that the complexity of strokes in our dataset does not require multiple Bézier curves per patch.

Results for both experiments are reported in Table 9.

| Stroke Length | Segments | Stroke Width | MSE |
|---|---|---|---|
| 3.0 | 1 | 0.4 | 0.0049 |
| 5.0 | 1 | 0.66 | **0.011** |
| 8.0 | 1 | 1.06 | 0.023 |
| 3.0 | 2 | 0.4 | 0.0052 |
| 5.0 | 2 | 0.66 | **0.017** |
| 8.0 | 2 | 1.06 | 0.023 |

*Table 9.* Impact of stroke length, number of segments per patch, and width.

## A.9. Analysis of Codebook Size

To better understand the influence of the codebook size $|V|$ on the reconstruction quality, we trained the VSQ module on the FIGR8 dataset using all the codebook sizes originally proposed in the Finite Scalar Quantization paper—namely 240, 1,000, 4,375 (used in our work), 15,360, and 64,000.

The results, presented in Table 10, reveal two key insights. First, increasing the codebook size leads to a substantial reduction in reconstruction error up to $|V| = 4,375$, indicating that a richer set of quantization centers significantly improves representation capacity in this range. However, beyond this threshold, further enlarging the codebook yields only marginal gains. This suggests that the added complexity and computational cost associated with very large codebooks may not be justified by the modest improvements in performance.

These findings support our choice of $|V| = 4,375$ as a balanced setting, offering strong performance with efficient resource usage.

| V | MSE |
|---|---|
| 240 | 0.0205 |
| 1000 | 0.0175 |
| 4375 | **0.0145** |
| 15360 | 0.0130 |
| 64000 | 0.0128 |

*Table 10.* MSE across varying codebook sizes.

## A.10. Im2Vec on Other Classes

We conducted a more in-depth analysis of the generative capabilities in Im2Vec after training on single subsets of FIGR-8, and compare the results with GRIMOIRE. We trained Im2Vec on the top-10 classes of FIGR-8: Camera (8,818 samples), Home (7,837), User (7,480), Book (7,163), Clock (6,823), Flower (6,698), Star (6,681), Calendar (misspelt as *caledar* in the dataset, 6,230), and Document (6,221). Table 11 compares the FID and CLIPScore with GRIMOIRE. Note that we train our model only once on the full FIGR-8 dataset and validate the generative performance using text-conditioning on the target class, whereas Im2Vec is unable to handle training on such diverse data. Despite Im2Vec appearing to obtain higher scores on several classes such as User or Document, a qualitative inspection reveals how the majority of the generated samples come in the form of meaningless filled blobs or rectangles. The traditional metrics employed in this particular generative field, based on the pre-trained CLIP model, react very strongly to such shapes in contrast to more defined stroke images. Qualitative samples are given in Table 19. We further observe a low variance in the generations when Im2Vec learns the representations of certain classes, such as star icons.

| Model | camera | | home | | user | | book | | clock | | cloud | | flower | | calendar | | document | |
|---|---|---|---|---|---|---|---|---|---|---|---|---|---|---|---|---|---|---|
| | FID | CLIP | FID | CLIP | FID | CLIP | FID | CLIP | FID | CLIP | FID | CLIP | FID | CLIP | FID | CLIP | FID | CLIP |
| Im2Vec (filled) | 9.21 | 27.86 | **3.48** | 26.85 | **2.12** | **28.92** | 7.18 | **27.26** | 6.12 | 26.38 | 17.43 | 24.38 | **6.61** | 25.42 | **4.5** | 27.26 | 12.19 | **28.65** |
| Im2Vec | 9.05 | 27.18 | 9.19 | 25.95 | 6.33 | 27.01 | 8.63 | 25.84 | **5.09** | 25.69 | 25.58 | 24.38 | 6.8 | 23.34 | 6.61 | 26.22 | 16.62 | 26.71 |
| GRIMOIRE | 6.74 | 29.81 | 7.16 | 27.16 | 5.45 | 26.81 | 6.65 | 27.1 | 7.22 | 26.32 | 6.78 | 24.96 | 10.27 | 22.00 | 5.57 | 26.23 | 4.08 | 27.96 |
| GRIMOIRE (+PC) | **5.77** | **30.22** | 7.6 | **27.41** | 4.38 | 27.18 | **5.8** | 27.24 | 6.79 | **26.45** | **6.05** | **25.51** | 9.37 | 22.46 | 5.09 | 26.41 | **3.81** | 28.21 |
| GRIMOIRE (+PI) | 7.5 | 29.46 | 7.44 | 27.01 | 5.95 | 26.85 | 6.79 | 27.08 | 7.63 | 26.12 | 7.09 | 24.73 | 9.97 | 22.04 | 5.87 | 25.98 | 4.21 | 27.89 |

*Table 11.* Quality of generations for GRIMOIRE and Im2Vec for the top-10 classes in FIGR-8.

## A.11. Qualitative Results of the Geometric Loss

The adoption of our geometric constraint improves the overall reconstruction error, which we attribute to the network being encouraged to elongate the stroke as much as possible. The results in Figure 15 show the effects on the control points of the reconstructed strokes from the VSQ. With the geometric constraint, the incentive to stretch the stroke works against the MSE objective, which results in an overall longer stroke and therefore in greater connectedness in a full reconstruction and an overall lower reconstruction error. We also present an example with an excessively high geometric constraint weight ($\alpha = 5$) demonstrating that beyond a certain threshold, the positive effect diminishes, resulting in degenerated strokes.

## A.12. Implementation Details

We use AdamW optimization and train the VSQ module for 1 epoch for Fonts and FIGR-8 and five epochs for MNIST. We use a learning rate of $\lambda = 2 \times 10^{-5}$, while the auto-regressive Transformer is trained for $\sim 30$ epochs with $\lambda = 6 \times 10^{-4}$. The Transformer has a context length of 512. Before proceeding to the second stage, we filter out icons represented by fewer than ten or more than 512 VSQ tokens, which affects 12.16% of samples. We use p-sampling for our generations with GRIMOIRE. Training the VSQ module on six NVIDIA H100 takes approximately 48, 15, and 12 hours for MNIST, FIGR-8, and Fonts, respectively; the ART module takes considerably fewer resources, requiring around 8 hours depending on the configuration. Regarding Im2Vec, we replace the Ranger scheduler with AdamW (Loshchilov & Hutter, 2017) and enable the weighting factor for the Kullback–Leibler (KL) divergence in the loss function to *0.1*, as it was disabled by default in the code repository, preventing any sampling. We train Im2Vec with six paths for 105 epochs with a learning rate of $\lambda = 2 \times 10^{-4}$ with early stopping if the validation loss does not decrease after seven epochs. Regarding the generative metrics, we utilized CLIP with a ViT-16 backend for FID and CLIPScore.

## A.13. Generative Scores with Completion

To evaluate if GRIMOIRE generalizes and learns to meaningfully complete previously unseen objects, we compare the CLIPScore and FID of completions with varying lengths of context. The context and text prompts are extracted from 1,000 samples of the test set of the FIGR-8 dataset. The results are shown in Table 12.

While GRIMOIRE can meaningfully complete unseen objects, the quality of these completions is generally lower than the generations under text-only conditioning. This is expected, as prompts in the test set are also encountered during training (the class names). The CLIPScore generally drops to its lowest point with the least amount of context and then recovers when more context is given to the model, which coincides with our qualitative observations that with only a few context strokes, GRIMOIRE occasionally ignores them completely or completes them in an illogical way, reducing the visual appearance.

| Model | Fonts | | FIGR-8 | |
|---|---|---|---|---|
| | FID | CLIP | FID | CLIP |
| GRIMOIRE (w/o context) | **1.67** | **28.64** | **3.58** | **27.45** |
| GRIMOIRE (+ 3 stroke context) | 2.78 | 27.25 | 4.65 | 25.31 |
| GRIMOIRE (+ 6 stroke context) | 3.16 | 27.25 | 5.46 | 25.54 |
| GRIMOIRE (+ 12 stroke context) | 2.95 | 27.57 | 6.04 | 25.85 |
| GRIMOIRE (+ 24 stroke context) | 2.25 | 28.12 | 6.05 | 26.39 |

*Table 12.* Generation quality of GRIMOIRE with different lengths of provided context on Fonts and FIGR-8. Post-processing is conducted for all setups. GRIMOIRE uses textual input for all generations.

### A.14. Domain Transfer Capabilities for Reconstruction

To validate how the strokes learned during the first training stage adapt to different domains, we use our VSQ module to reconstruct Fonts after training on FIGR-8, and vice versa. Figure 16 provides a qualitative example for each setting. Despite the loss value for each image being around one order of magnitude higher than the in-domain test-set (MSE$\approx 0.05$), the VSQ module uses reasonable codes to reconstruct the shapes and picks curves in the correct directions. Straight lines end up being the easiest to decode in both cases.

### A.15. Codebook Usage for Strokes

As described in Section 3.1, for FSQ, we fixed the number of dimensions of the hypercube to 5 and set the individual number of values for each dimension as $L = [7, 5, 5, 5, 5]$ for a total codebook size of $|B| = 4,375$. In this section, we wish to share some interesting findings about the learnt codebook. For this, we shall use the VSQ trained on FIGR-8 with $n_{code} = 1$, $n_{seg} = 2$, a maximum stroke length of $3.0$, and the geometric constraint with $\alpha = 0.2$.

After training the VSQ on FIGR-8, we tokenize the full dataset. The resulting VQ tokens stem from 60.09% of the codebook, while 39.91% of the available codes remained unused. The ten most used strokes make up 41.24% of the dataset, while the top 24 and 102 strokes make up roughly 50% and 75%, respectively. These findings indicate that for these particular VSQ settings, one could experiment with smaller codebook sizes.

To balance out the stroke distribution, one could use a different subset of FIGR-8. Currently, the classes "menu", "credit card", "laptop", and "monitor" contribute the most to the stroke imbalance, with 26%, 24.3%, 24.05%, and 23.8% of their respective strokes being the most frequent horizontal one in Table 13.

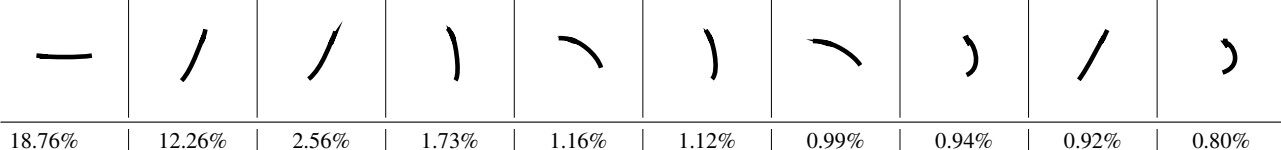

| 18.76% | 12.26% | 2.56% | 1.73% | 1.16% | 1.12% | 0.99% | 0.94% | 0.92% | 0.80% |

*Table 13.* Top ten most used strokes of the VSQ module trained on icons and their relative occurrences in our subset of FIGR-8.

### A.16. Average Strokes in Codebook

In Section A.15, we show the ten most used strokes of our trained VSQ, but after inspecting the full codebook, we notice how neighboring codes often express very similar strokes. Therefore, to visualize the codebook more effectively, we plot mean and minimum reductions of the full codebook in Figure 17. Additionally, we tokenize the full FIGR-8 dataset and plot the same reductions in Figure 18 to show the composition of the dataset.

### A.17. Qualitative Results – Reconstruction

In Table 14 and Table 15, we provide several qualitative examples of vector reconstructions using Im2Vec and our VSQ module on the Fonts and FIGR-8 datasets, respectively. We fill the shapes of the images when using Im2Vec, since the model creates SVGs as series of filled circles and would not be able to learn from strokes with a small width. Im2Vec does not converge when trained on the full datasets, whereas it returns some approximate reconstruction of the input when only a

single class is adopted. In contrast, the VSQ module generalizes over the full dataset.

## A.18. Qualitative Results – Generation

In this section, we provide qualitative examples of our reconstruction and generative pipeline, and compared those with Im2Vec. Table 16 reports a few examples of icons generated with GRIMOIRE using only text-conditioning on classes. In Table 17 we report some generations for MNIST. In Table 18, we report generative results for Fonts. Thanks to the conditioning, we can generate upper-case and lower-case glyphs in bold, italic, light styles, and more. As can be seen in the table, GRIMOIRE also learns to properly mix those styles only based on text. Finally, in Table 19, we report some generative results on icons and Fonts for Im2Vec on a single class dataset. The results show how the pipeline typically fails to produce meaningful or sufficiently diverse samples.

## A.19. Glossary of notation

Table 20 reports a description of all the important notations used in this work.

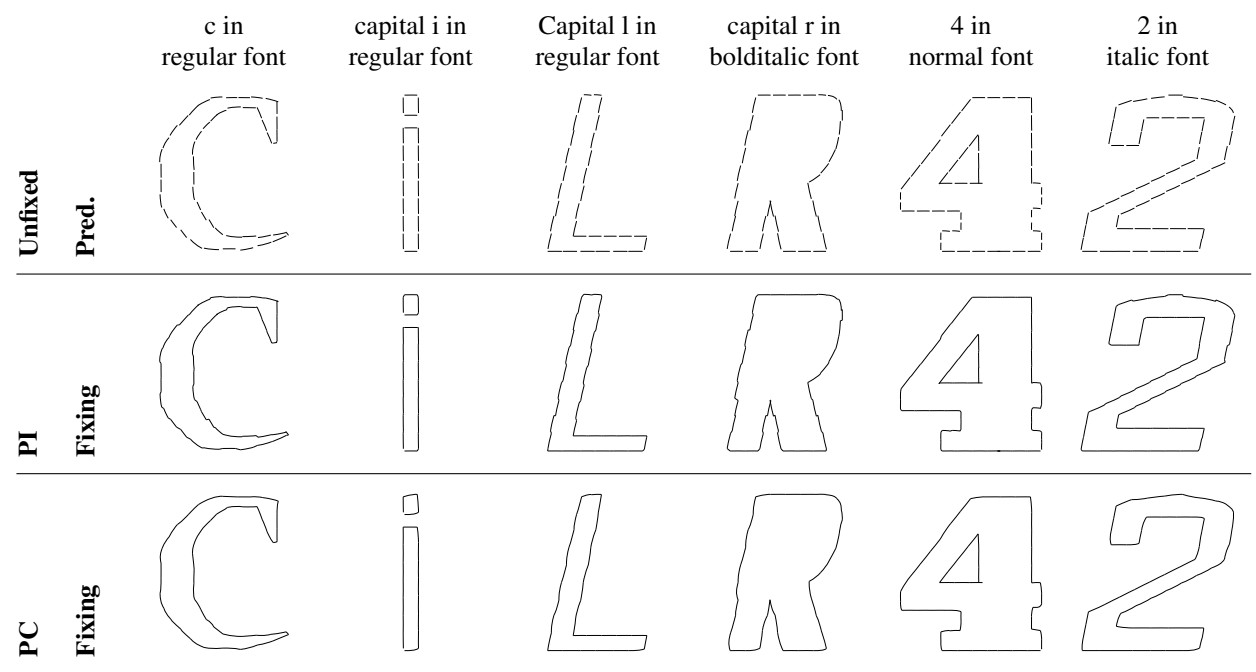

*Figure 14.* Some examples of text-conditioned glyph generation from GRIMOIRE. The first row shows the unfixed model predictions, the second and third rows depict the final outputs with two different post-processing techniques.

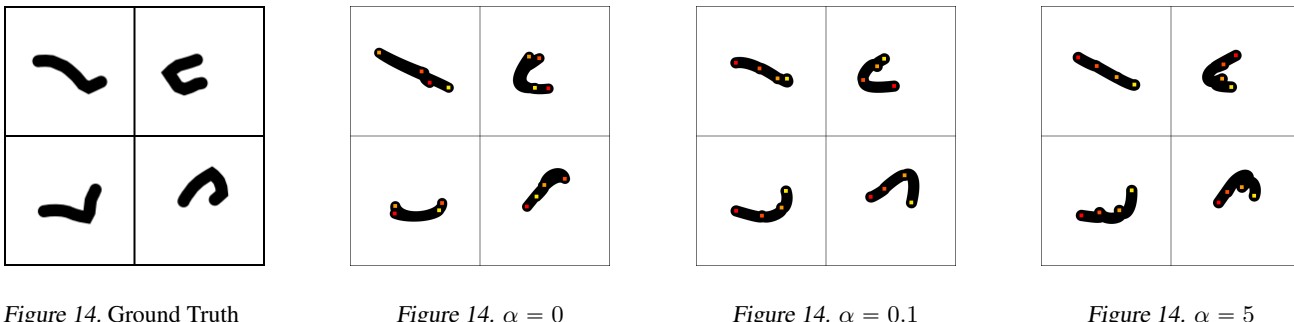

*Figure 14.* Ground Truth         *Figure 14.* $\alpha = 0$         *Figure 14.* $\alpha = 0.1$         *Figure 14.* $\alpha = 5$

*Figure 15.* Samples from the test set when training the VSQ module with and without our geometric constraint. Each stroke consists of two cubic Bézier segments. Embedded within each stroke, the red dots mark the start and end points, while the green and blue dot pairs are the control points of each segment.

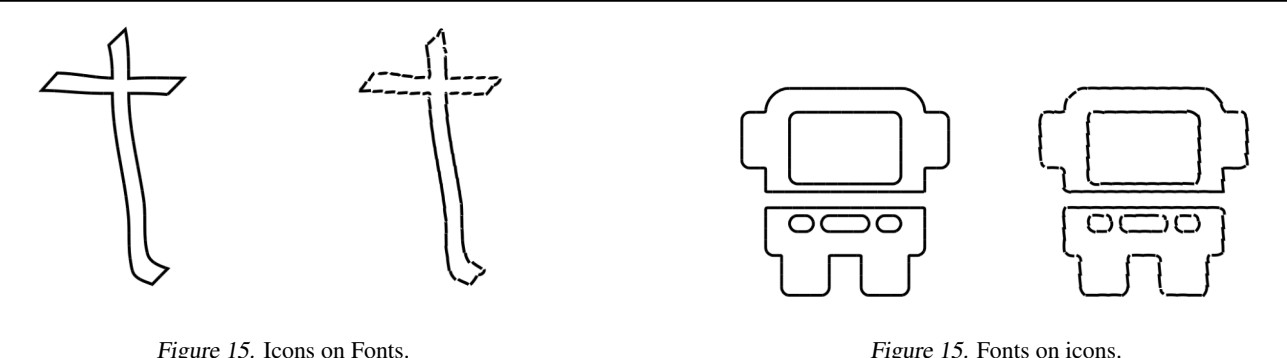

*Figure 15.* Icons on Fonts.                          *Figure 15.* Fonts on icons.

*Figure 16.* Qualitative zero-shot reconstructions from the test-set of FIGR-8 and Fonts after training the VSQ module solely on the respective other dataset.

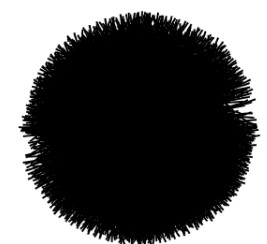

*Figure 16.* codebook mean strokes

*Figure 16.* all codebook strokes

*Figure 17.* Different reductions of all 4,375 strokes from the VSQ codebook. The model seems to have learned an expressive codebook-decoder mapping as the figure on the left shows a smooth and evenly distributed stroke profile and the figure on the right displays strokes in almost every direction.

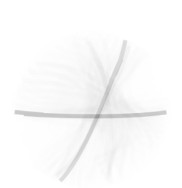

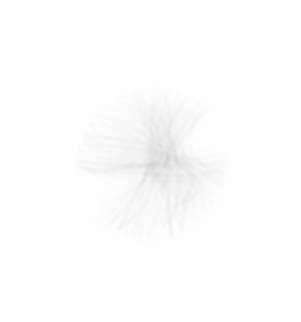

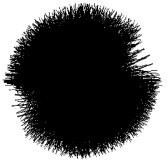

*Figure 17.* FIGR-8 mean strokes

*Figure 17.* FIGR-8 mean strokes excluding top ten strokes

*Figure 17.* all FIGR-8 strokes

*Figure 18.* Different reductions of all strokes from the tokenized FIGR-8 dataset. The visualization on left shows the dominance of the two most occurring strokes, the middle shows that the distribution of strokes is skewed. The missing 39.91% of strokes are also visible in the right figure, where certain diagonal strokes that are available in the codebook are never used.

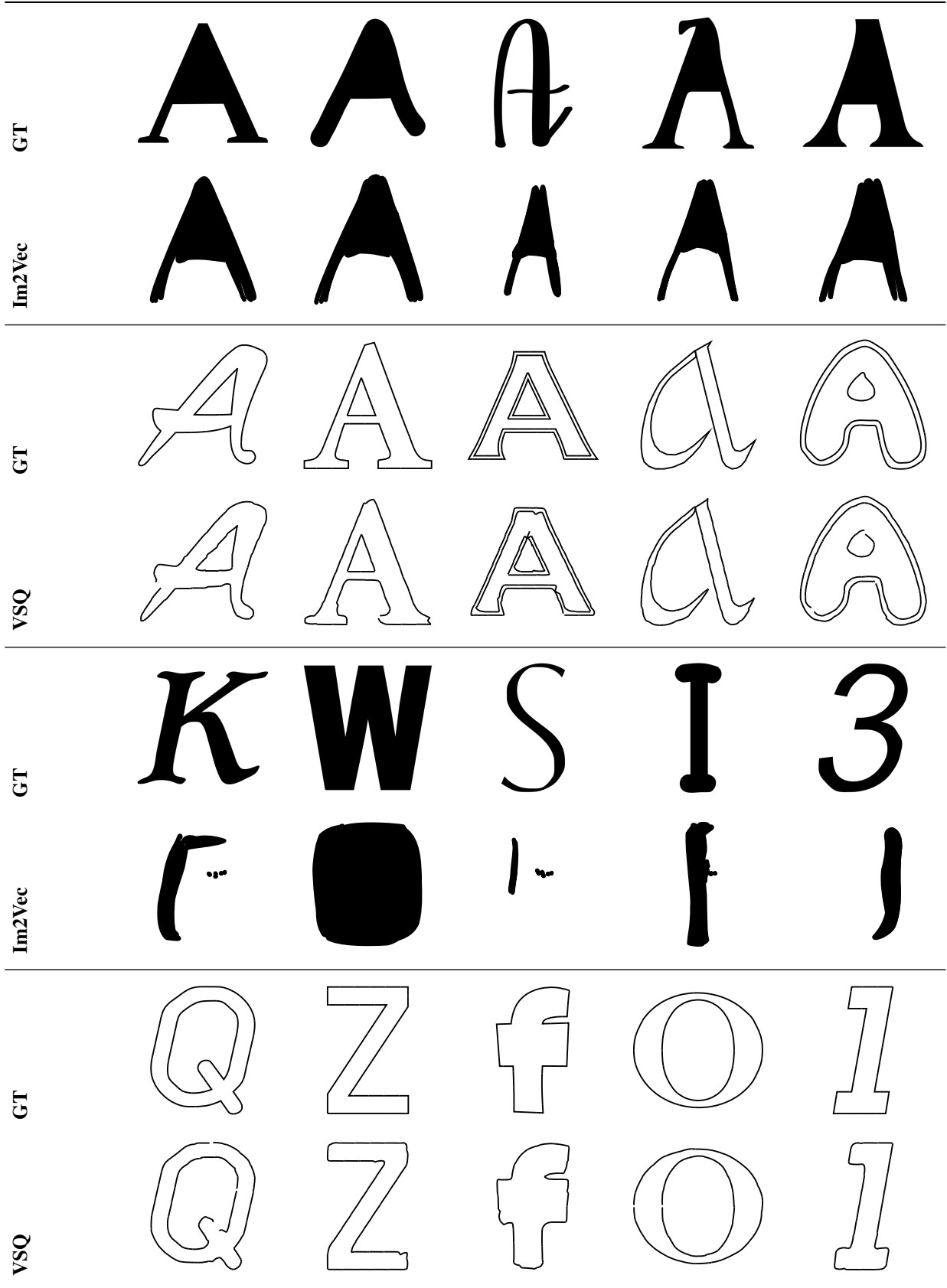

*Table 14.* Examples of various reconstructions of our VSQ module after training on Fonts compared to reconstructions of Im2Vec trained on the letter "A" (first row) and Im2Vec trained on the full Fonts dataset (third row).

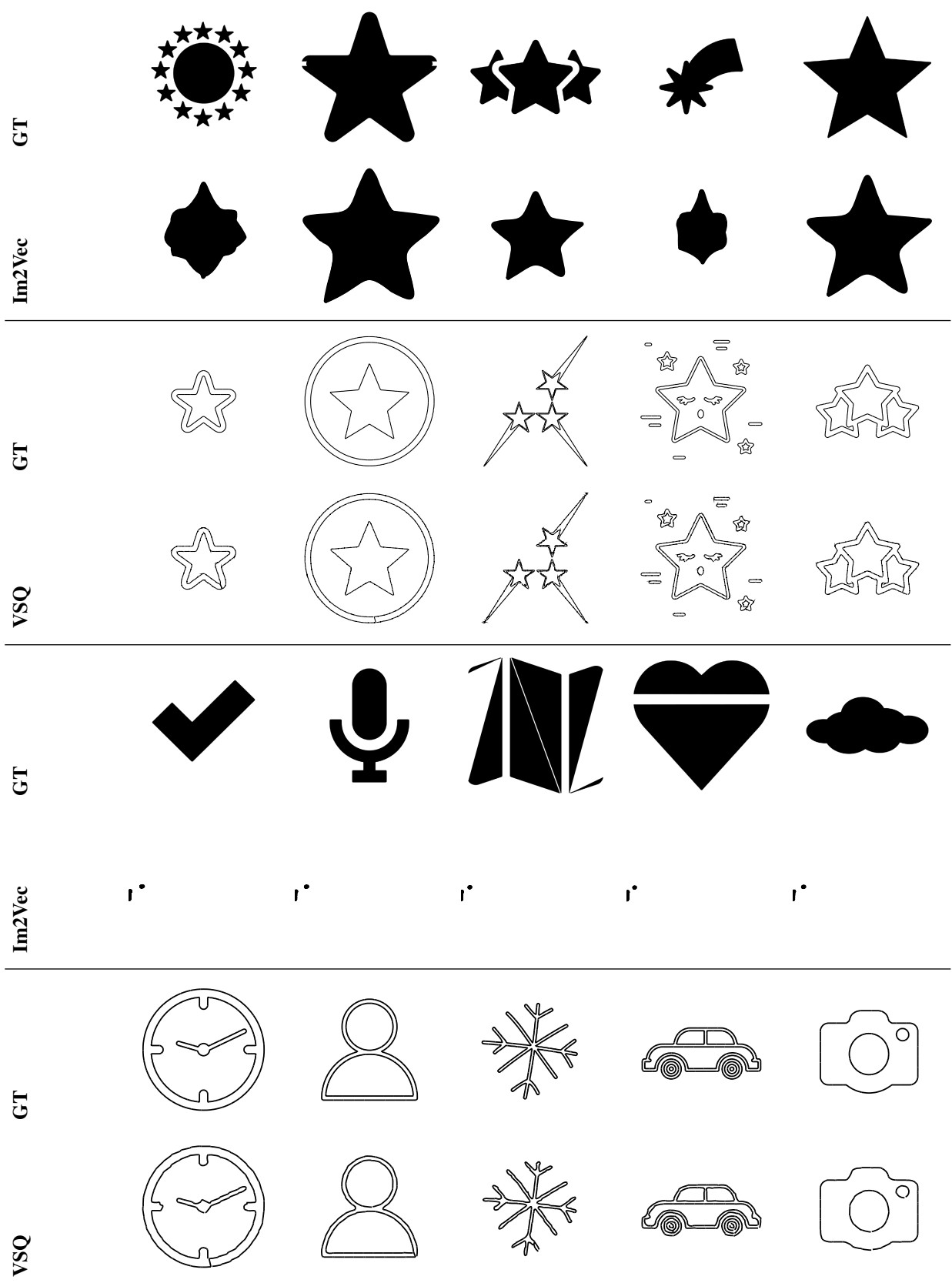

*Table 15.* Examples of various reconstructions of our VSQ module after training on icons compared to reconstructions of Im2Vec trained on one class (first row) and Im2Vec trained on the full dataset (third row).

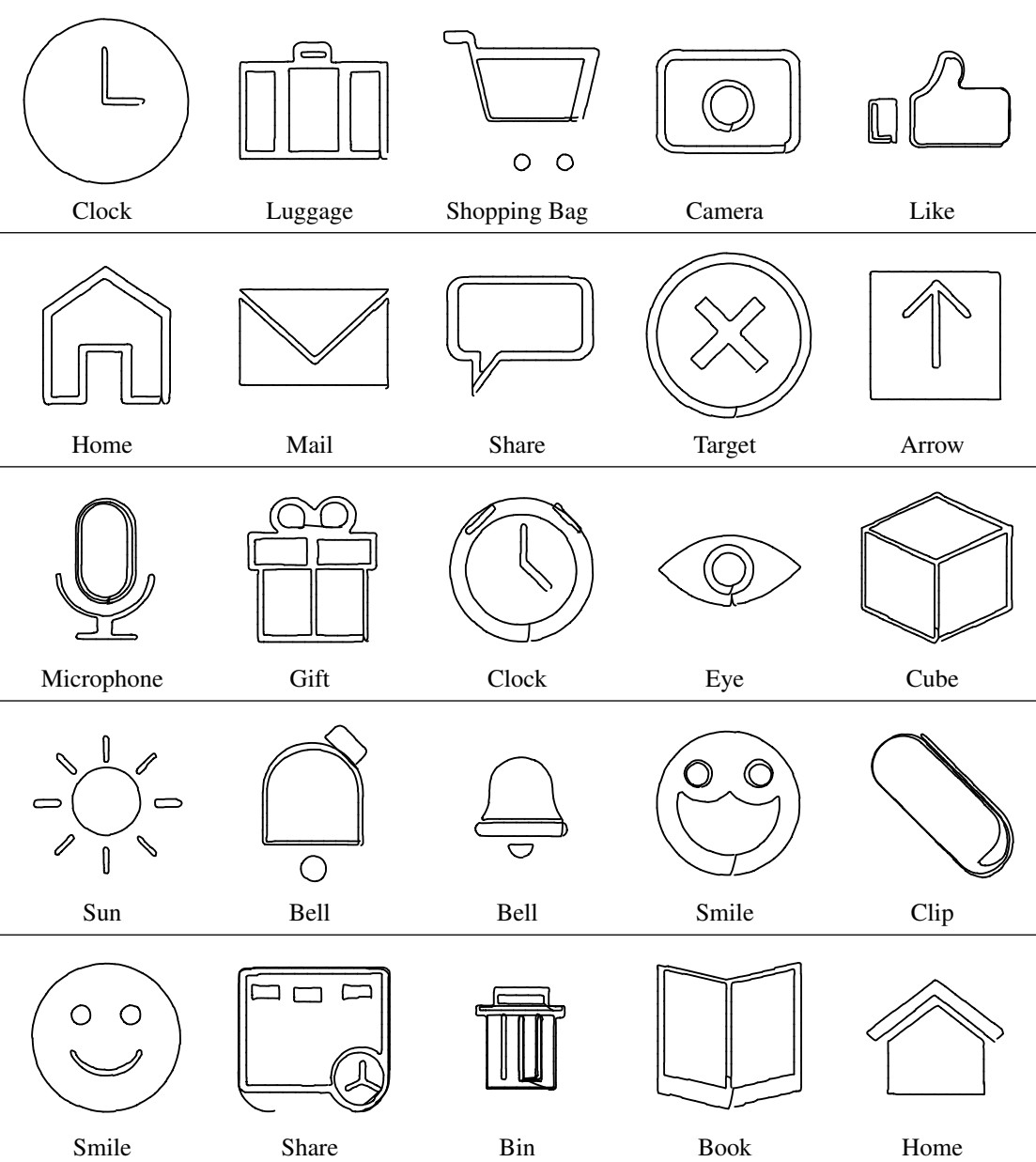

| | | | | |
|---|---|---|---|---|
| Clock | Luggage | Shopping Bag | Camera | Like |
| Home | Mail | Share | Target | Arrow |
| Microphone | Gift | Clock | Eye | Cube |
| Sun | Bell | Bell | Smile | Clip |
| Smile | Share | Bin | Book | Home |

*Table 16.* Examples of various samples generated with GRIMOIRE after training on icons, using only text conditioning.

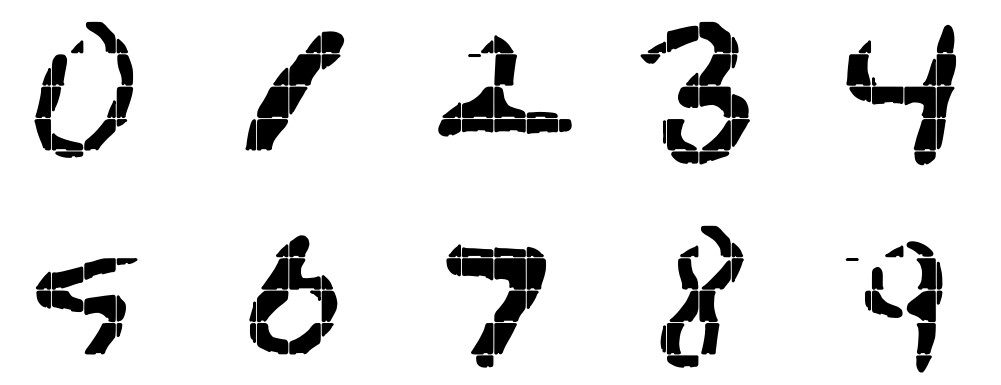

*Table 17.* Examples of a samples generated with GRIMOIRE for each digit of the MNIST dataset.

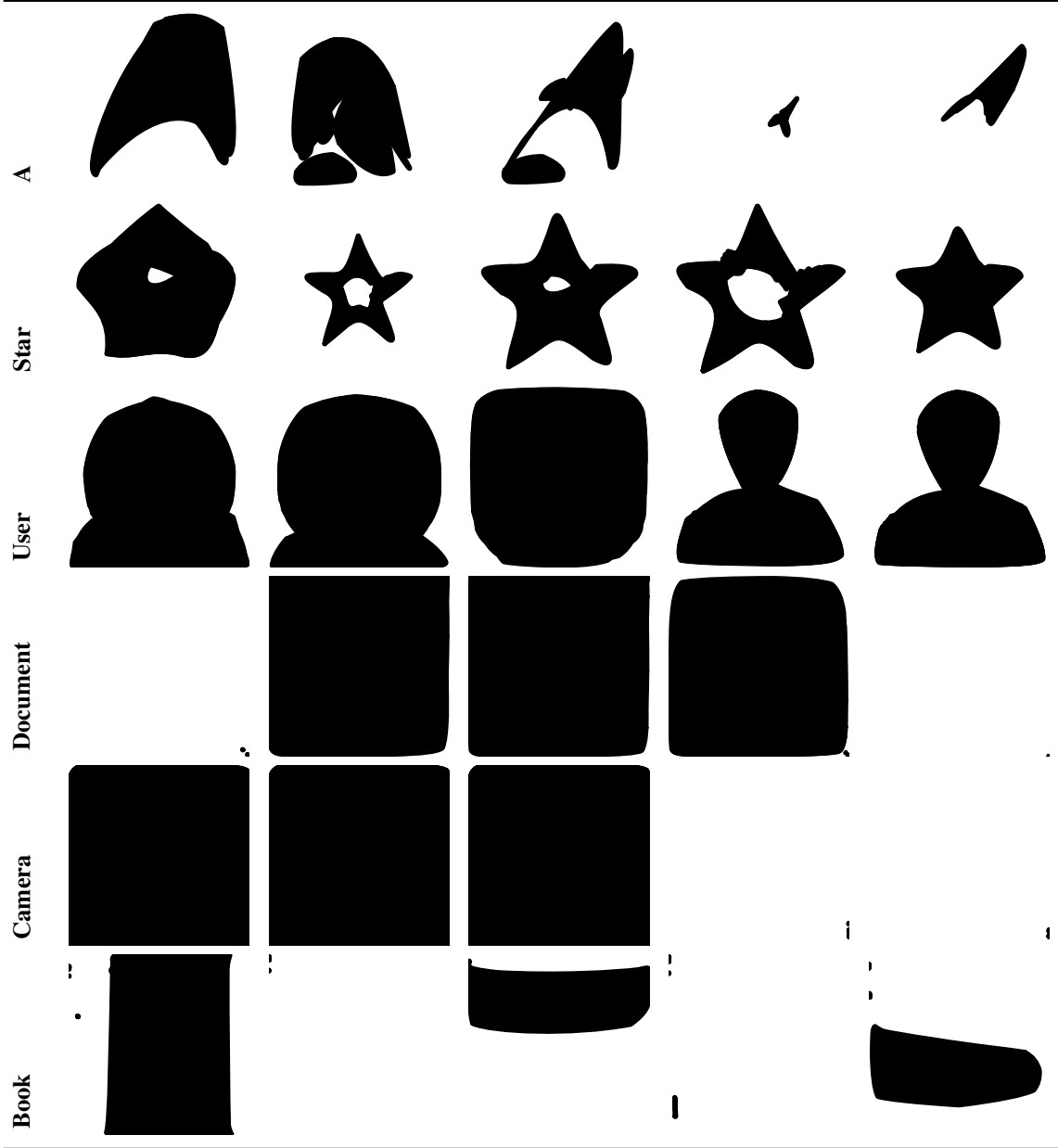

*Table 18.* Examples of filled samples generated with Im2Vec after training the model on specific classes of the dataset. For most classes, Im2Vec could not capture the diversity of the data and failed to meaningfully converge.

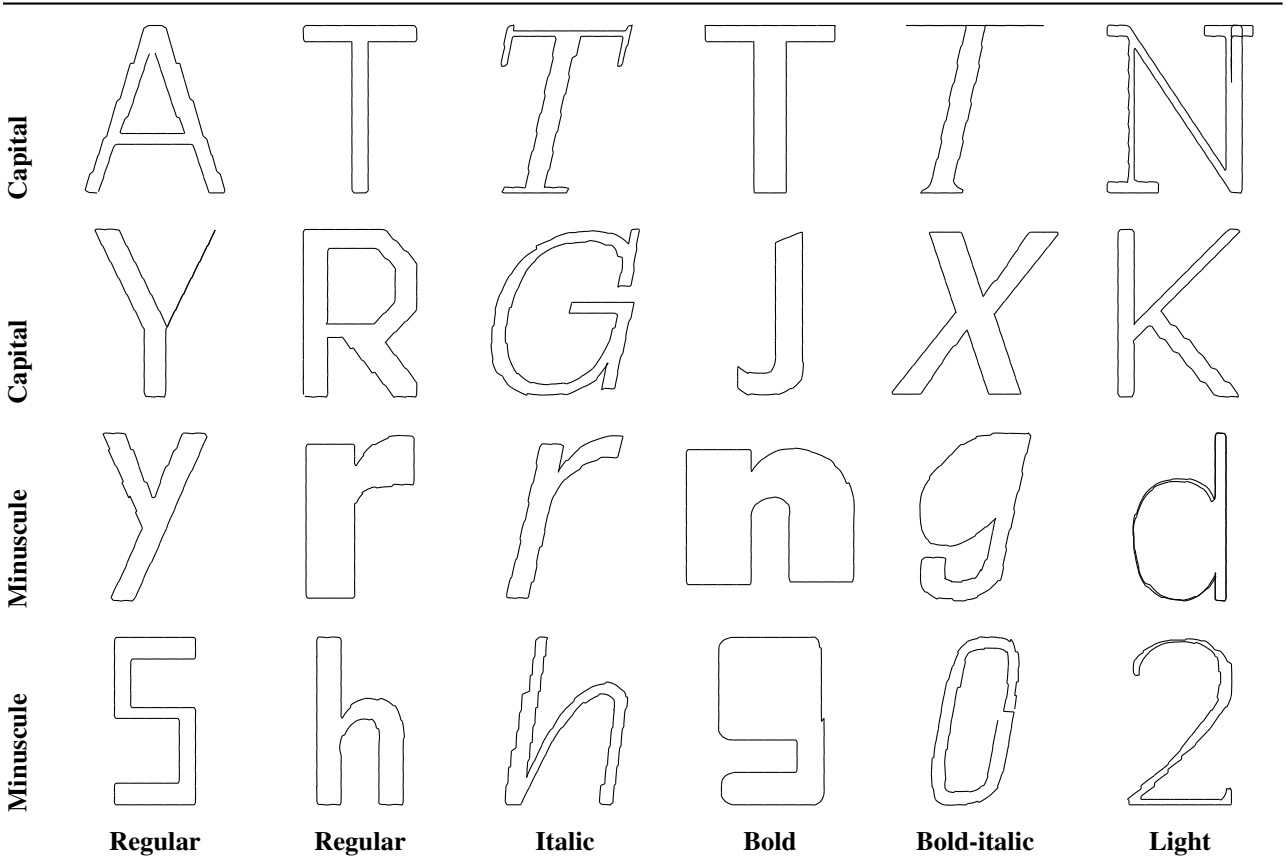

*Table 19.* Examples of various samples generated with GRIMOIRE after training on Fonts, using only text conditioning.

*Table 20.* Glossary of all notations.

| | |
|---|---|
| $E$ | Network encoder |
| $D$ | Network decoder |
| $I$ | Image from the dataset |
| $V$ | Codebook |
| $v$ | Codes from the codebook |
| $L$ | Set of values per dimension of our codebook |
| $l$ | Single dimensional value |
| $q$ | Number of dimensions of the codebook |
| $\mathbf{S}$ | Series of patches |
| $s$ | Single patch |
| $C$ | Color channels |
| $n$ | Number of patches |
| $\Theta$ | Set of discrete coordinates |
| $\theta$ | Single coordinate pair |
| $\mathcal{Z}$ | Latent space |
| $d$ | Dimension of latent |
| $z$ | Latent embedding |
| $\hat{s}$ | Predicted patch |
| $\nu$ | Number of segments |
| $P$ | Set of points |
| $p$ | Point pair |
| $\rho$ | Euclidian distance |
| $\Phi$ | Neural network |
| $\xi$ | Number of codes |
| $T$ | Text description |
| $\mathcal{T}$ | Tokenized description |
| $\tau$ | Text tokens |
| $t$ | Number of text tokens |

