# OpenReview forum: "Vector Grimoire: Codebook-based Shape Generation under Raster Image Supervision"
_ICML.cc/2025/Conference — ICML 2025 poster_

### Official Review · Reviewer_sRe3 · 2025-03-12

**Overall Recommendation:** 2

**Summary:**

This work focuses on SVG generation. Its model consists of two modules: a visual shape quantizer learns to map raster images onto a discrete codebook by reconstructing them as vector shapes, and an auto-regressive Transformer model jointly learns the distribution over shape tokens, positions and textual descriptions.

**Claims And Evidence:**

There is no comparison with IconShop on FIGR-8. This work does not use ground-truth paths as the supervision while Iconshop uses ground-truth paths. If there is no such direct comparison, it is impossible to identify the contribution of the proposed work: a) it achieves better performance than the one uses ground-truth path. If this is the case, the contribution is very significant, i.e., changing the traditional paradigm of SVG generation; or b) it does not achieve better performance than the one uses ground-truth path. If that is the case, the contribution simply lies on avoiding using ground-truth paths.

**Essential References Not Discussed:**

The advanced SDS-based methods are not discussed [1][2][3]

[1] https://arxiv.org/pdf/2312.16476

[2] https://arxiv.org/pdf/2405.10317

[3] https://arxiv.org/pdf/2411.16602 (optional as it is too close to the submission deadline)

**Experimental Designs Or Analyses:**

In the comparison to the SDS-based methods, the chosen baselines are not the most advanced ones [1][2][3] and the results of SDS-based methods look much worse than what are reported in the original paper. I suggest the authors make more explanation for it.

[1] https://arxiv.org/pdf/2312.16476

[2] https://arxiv.org/pdf/2405.10317

[3] https://arxiv.org/pdf/2411.16602 (optional as it is too close to the submission deadline)

**Methods And Evaluation Criteria:**

There is no qualitative comparison between baselines and the proposed method. The quantitative comparison makes sense.

**Other Comments Or Suggestions:**

no additional comments

**Other Strengths And Weaknesses:**

no additional comments.

**Questions For Authors:**

Does the extension to support stroke width and color prediction mentioned in Sec 5.3 requires re-training of the proposed two modules?

**Relation To Broader Scientific Literature:**

It introduces a new paradigm of training SVG generation model which does not rely on SVG data (which requires lots of data for training) and is not SDS-based (which can be very slow). The idea of splitting the image to patches in VSQ stage is smart.

**Theoretical Claims:**

There are no theoretical claims.

---

> ### Author Rebuttal · Authors · 2025-03-30
>
> We sincerely appreciate the reviewer's time and thoughtful feedback on our manuscript.
> Given the time constraints of this rebuttal, we have focused on addressing the major concerns as follows.
>
>
> ---
> ## **Vector-based baselines**
>
> We have extended our analysis to two vector-supervised methods – **DeepSVG and IconShop** – training them on the same FIGR-8 data used for Grimoire. Unlike Grimoire, DeepSVG supports conditioning only on class identifiers, therefore we assigned a unique identifier to each class in FIGR-8.
>
> Upon suggestion of reviewer `f1Fu`, **we have also finetuned Llama 3.2** on FIGR-8 with minimal data preprocessing. We believe this to be an insightful analysis that **shows how tailored tokenization pipelines and extensive data preprocessing are necessary for other vector-supervised models to perform effectively.**
>
> Despite raster and vector data providing very different supervising signals, we believe that this analysis ultimately helps better position our method.
>
> **Llama**. We fine-tuned Llama (instruction tuning) for three days on eight H100 GPUs. Minimal preprocessing includes rounding up the path coordinates to integer values. Upon inspection, this did not affect the quality of the image. We use the original chat template and included special tokens to delimit the SVG code. **The performance at inference appears very poor**. The model predicts the most recurrent patterns in the dataset, resulting mainly in circular artifacts. **The SVG syntax is, however, correct** most of the time, allowing rendering.
>
> **DeepSVG**. We train DeepSVG using the official training script. The model converges within a few hours, but **the results are also not good**, yielding the lowest CLIPScore and FID among all models.
>
> **IconShop**. We also re-trained the original IconShop model on the subset of FIGR-8 used in Grimoire. In this case, **the performance of the model is comparable to Grimoire**, resulting in slightly better CLIPscore and FID.
>
> All results are reported in the table below.
>
> | Model     | CLIPScore | FID   | Conditioning | Supervision |
> |-----------|-----------|-------|--------------|-------------|
> | DeepSVG   | 22.10     | 58.03 | Class        | Vector      |
> | Llama 3.2 | 25.45     | 38.93 | Prompt       | Vector      |
> | Grimoire  | 29.00     | 0.64  | Prompt       | Raster      |
> | IconShop  | 31.18     | 0.40  | Prompt       | Vector      |
>
> ---
> ## **Additional SDS-based methods**
>
>
> We have added two more recent SDS-based methods (**SVGDreamer and Chat2SVG**) to our qualitative analysis for the final version of our manuscript. We have also included a **new quantitative analysis of all these models.**
>
> The point of the results in section 4.5 is not comparing general generative capabilities but highlighting that besides the aesthetically pleasing results, **this family of models falls short in representing a specific target domain** and provides no way to be extended to new data.
>
> Making this analysis quantitative is not straightforward. FID score between image distribution  is reliable on thousands of samples, but the computational cost of SDS-based models requires up to hours for a few samples (e.g. SVGDreamer) or uses costly proprietary models (e.g. Chat2SVG).
>
> We have hence used the PSNR of 20 generated samples from all models.
> **The results highlight how all models fall short on our dataset distribution.**
>
> | Class         | Model         | Average PSNR (dB) |
> |---------------|---------------|-------------------|
> | **User**      | CLIPdraw      | 28.68             |
> |               | **Grimoire**  | **45.19**         |
> |               | VectorFusion  | 36.62             |
> |               | Chat2SVG      | 37.62             |
> |               | SVGDreamer    | 34.53             |
> | **Heart**     | CLIPdraw      | 28.54             |
> |               | **Grimoire**  | **45.66**         |
> |               | VectorFusion  | 38.54             |
> |               | Chat2SVG      | 37.88             |
> |               | SVGDreamer    | 34.44             |
>
> ---
> ## **Other questions and observations**
> > Does the extension to support stroke width and color prediction mentioned in Sec 5.3 require re-training of the proposed two modules?
>
> It does require retraining the VSQ. This is also the case for vector-supervised methods.
>
> However, a significant difference is that our VSQ prediction heads are very easy to develop or enable and require no further changes, whereas vector-supervised methods like IconShop require redesigning their tokenizer and reprocessing all the data.
>
> Finally, if the VSQ must be retrained with some additional modules (e.g., stroke color and width), the rest of the network can be reloaded, speeding up convergence.
>
> > "the contribution simply lies on avoiding using ground-truth paths."
>
> The main novelty of our work is providing a text-to-SVG framework which learns from data where the vector paths **do not exist**. This is different from simply avoiding existing vector datasets.

---

> > ### Comment · Reviewer_sRe3 · 2025-04-04
> >
> > Thanks for the feedback. I would like to learn more comments from other reviewers before making a final decision. I currently have two concerns. First, while I acknowledge that IconShop utilizes SVG path data while the proposed method does not, IconShop still outperforms the proposed approach. This raises questions about the necessity and significance of the proposed method. Second, a key advantage of SDS-based methods is their ability to operate independently of specific data distributions by leveraging prior knowledge from a pre-trained model. In contrast, the proposed method lacks this flexibility.

---

### Official Review · Reviewer_dhph · 2025-03-13

**Overall Recommendation:** 3

**Summary:**

This paper introduces a text-guided SVG generation model, i.e., GRIMOIRE, using only raster image supervision. The SVG generation task is formulated as the prediction of a series of individual shapes and positions. The experiments demonstrate the effectiveness of the proposed method.

## update after rebuttal

I appreciate the authors' clarifications. Most of my concerns have been addressed by the rebuttal. I would lean to keep my score by involving the additional evaluations and discussions in the revised version.

**Claims And Evidence:**

The discussion of the existing SDS-based methods is not very clear. Sec. 2.1 only describes several related works without discussing the difference between the existing works and the proposed methods. Sec. 5.4 only shows visual comparisons with the existing works. It would be better to involve the analysis in the paper, rather than supp.

**Essential References Not Discussed:**

No.

**Experimental Designs Or Analyses:**

The current evaluation is not thorough. Only Im2vec is used for comparison. To make the result more convincing, additional existing SVG generation works should be compared and discussed.

**Methods And Evaluation Criteria:**

The proposed method is a reasonable and effective solution to SVG generation.

**Other Comments Or Suggestions:**

It would be better to reorganize the structure of section 5. The first paragraph states two aspects of the result, but there are four subsections here.

**Other Strengths And Weaknesses:**

The overall pipeline is a reasonable solution to perform SVG generation. The experiments demonstrate the effectiveness of the proposed method.

**Questions For Authors:**

Are there any quantitative evaluations for text-conditioned SVG generation?

**Relation To Broader Scientific Literature:**

The proposed method enhances the generation quality over existing raster-supervised SVG models and enables flexible text-conditioned SVG generation.

**Theoretical Claims:**

Yes.

---

> ### Author Rebuttal · Authors · 2025-03-30
>
> We sincerely thank you for taking the time to review our manuscript and providing valuable feedback.
>
>
> ---
> ## **Discussion on SDS-based Methods**
>
> In the final version of the manuscript, we plan to more clearly highlight the differences between SDS approaches and Grimoire at the end of section 2.1 as follows:
>
> > “These methods do not involve training for the vector generation process and instead rely on models trained on raster images for other tasks, making them difficult to extend to new data.”
>
> Regarding this point, we have also:
> 1. incorporated two more recent SDS-based methods, **SVGDreamer** and **Chat2SVG**, and will include additional qualitative results in the final version of the manuscript.
> 2. Added a quantitative comparison of these methods. This analysis has been included in our response to Reviewer `sRe3`.
>
>
> We used **PSNR** to evaluate some generated samples from all the SDS-based models, highlighting how these methods fall short when compared to a specific dataset. We chose not to use **FID**, as it requires a large number of samples to be statistically significant, which was not feasible within the constraints of SDS methods: slow generations e.g. SVGDreamer, expensive inference e.g. Chat2SVG based on Claude APIs.
>
> ---
> ## **Other Questions**
>
> > Are there any quantitative evaluations for text-conditioned SVG generation?
>
> We have used CLIPScore and FID for text-conditioned generation across the paper. As mentioned above, we have now also added PSNR for the comparison with SDS-based methods.
>
> > The first paragraph states two aspects of the result, but there are four subsections here.
>
> Thank you for your suggestion on improving the writing in Chapter 5. We plan to revise the beginning of the chapter to more clearly outline its subsections.

---

### Official Review · Reviewer_f1Fu · 2025-03-13

**Overall Recommendation:** 3

**Summary:**

The authors propose a SVG generative model GRIMOIRE which can be conditioned on a text prompt or a partially completed SVG. The primary innovation in the paper is training a VQ-VAE which tokenizes patches of rasterized SVGs into discrete tokens which crucially can be reconstructed into SVG primitives (primarily bezier curves). This tokenizer can be trained end-to-end on purely raster images by a clever use of a differentiable renderer (DiffVG) in the VQ-VAE decoder which enables backpropagting through the L2 pixel space reconstruction loss into and through the now differentiable continuous SVG parameters. Once such a tokenizer has been trained the authors use it to tokenize MNIST, Fonts and FIGR-8 dataset and train a standard autoregressive transformer using a next token prediction loss to create an auto-regressive generative model for SVGs. The primary baseline for comparison is Im2Vec.

**Claims And Evidence:**

Yes.

**Essential References Not Discussed:**

LIVE is definitely an important missing reference: https://arxiv.org/abs/2206.04655. There are also a series of follow-up papers building on top of LIVE.

Likewise StarVector: https://arxiv.org/abs/2312.11556

**Experimental Designs Or Analyses:**

Checked.

**Methods And Evaluation Criteria:**

- The benchmark datasets seem reasonable.
- However in my opinion there are a number of missing baselines. In particular for the reconstruction experiments in Tables 1 and 2 that validate the VQ-VAE I would suggest also including LIVE (https://arxiv.org/abs/2206.04655) a seminal paper in the area of SVG reconstruction of raster images. StarVector: https://arxiv.org/abs/2312.11556 would also be an appropriate baseline for the generation experiments.
- Additionally, as a simple baseline that would be very insightful, fine-tuning a standard LLM e.g. Llama on the Fonts and FIGR-8 datasets, which IIUC have the underlying SVG code available, using minimal preprocessing of the SVG code and using the standard Llama tokenizer directly on the SVG code. This will validate or invalidate the need for such a complex tokenization method and the claims in the paper about the need for more rasterized training data and complex pre-processing required in order to train a SVG generation model directly on the SVG code.
- Beside all quantitative results I would have also liked to have seen the average SVG codelength of all methods (GRIMOIRE and Im2Vec) for each dataset. When available, the average SVG codelength in the ground truth dataset would also be helpful. It is currently very difficult to understand how fair the comparison between the methods is, without some way to know how much compression each method is doing. I have a suspicion given the number of bezier curves used to represent each, relatively simple, image patch that GRIMOIRE may be relatively uncompressed and verbose.
- No measure of significance or error bars are provided on any of the quantitative results.

**Other Comments Or Suggestions:**

N/A

**Other Strengths And Weaknesses:**

Strengths:

- I appreciate that the authors will make their code open source.
- The use of DiffVG to enable learning a discrete codebook for SVG primitives, and the fact that this enables training the tokenizer on purely rasterized images is very clever and a very nice contribution of the paper.

Weaknesses:
- The only structural primitive available to the VQ-VAE decoder are Bezier curves. SVG itself supports other primitives e.g. <circle>, <line>, etc. which may be more appropriate, interpretable and compressed representations of certain image patches. But having to make a discrete choice between different primitives would make the decoder non-differentiable meaning that other than for stylization (stroke width, color) all structure in the SVG must be represented as relatively uncompressed, uninterpretable paths.
- The need for specialized path extraction methods on a per dataset basis as shown in Figure 3 is a major weakness of the method. Surely a more general patch extraction method would have been possible to employ without a significant degradation in quality? It would have also enabled training a single joint codebook and generative model for all datasets enabling potential transfer learning and greatly enhancing the generality of the method.

**Questions For Authors:**

N/A

**Relation To Broader Scientific Literature:**

A very nicely written related work section is provided.

**Theoretical Claims:**

N/A

---

> ### Author Rebuttal · Authors · 2025-03-30
>
> We sincerely appreciate your time and thoughtful feedback on our manuscript.
>
> Given the time constraints of this rebuttal, we have focused on addressing the major concerns as follows.
>
> ---
> ## **Code Length of the SVG**
>
> For Im2Vec, the number of paths and control points per path is fixed at eight and ten, respectively.
> **Grimoire dynamically adjusts the number of strokes based on the target complexity.**
>
> After inspecting over 15,000 generated SVGs across all FIGR-8 classes, we found that **the average number of paths is 95**.
>
> **We have uploaded samples and reconstructions for the “user” class for both models to our anonymous repository**. For the reconstructions we included the original ground truth for reference.
>
> For single-target reconstructions, Im2Vec tends to overlap strokes around the outline, whereas for more complex targets, the strokes collapse. We encourage the reviewer to directly inspect the SVG code length on the anonymous GitHub Repository: [Link](https://github.com/anon-papercode/15874/tree/main/Showcase).
>
> We found this analysis insightful and plan to incorporate it into the results section in the final version of the manuscript.
>
> ---
> ## **Additional Baselines**
>
> **Thank you for suggesting finetuning a standard LLM with minimal preprocessing.** This was an insightful suggestion, especially given that the tokenization pipelines of other vector-supervised models (e.g., IconShop) are a considerable limitation.
>
> **We have fine-tuned LLaMA 3.2 on the same FIGR-8 subset used for Grimoire with minor preprocessing** and also added comparisons with other vector-supervised models using their respective tokenization pipelines (DeepSVG, IconShop).
>
> This analysis has been included in our response to reviewer `sRe3`.
>
> ---
> ## **Missing References**
>
> We appreciate all reviewers for highlighting important missing references, such as LIVE.
> We plan to expand the related work section to incorporate all suggested papers. Specifically, we will:
>
> 1. **Before L87:** Introduce vector-supervised methods that predate the LLM era, citing **DeepSVG, Google-Fonts, and DeepVecFont**.
> 2. **Immediately after:** Among the LLM-based approaches, explicitly mention **StarVector** and **Chat2SVG**.
> 3. **At the end of the section:** Where we discuss SDS-based methods, include **SVGDreamer** and dedicate a small paragraph to neural implicit representations, citing **NiVel, Text-to-Vector Generation with Neural Path Representation, and NeuralSVG**.
>
> While these works differ significantly in methodology, they address similar problems to Grimoire and will be appropriately cited.
>
> We hope this revision sufficiently addresses concerns regarding missing references.

---

> > ### Comment · Reviewer_f1Fu · 2025-04-07
> >
> > I thank the author's for their response to my review. In particular I appreciate the addition of the LLaMA baseline, the inclusion and results of which I regard as a positive addition to the paper. I also appreciate adding the average SVG code-lengths (in terms of number of paths) which demonstrated, as I feared, that the Grimoire code is relatively complex and verbose.
> >
> > Balancing these two factors, I maintain my original score. The paper is interesting and a useful addition to the literature with some significant disadvantages.

---

> > > ### Author Response · Authors · 2025-04-07
> > >
> > > Thank you for your positive comments and for appreciating our work!
> > >
> > > We would like to kindly offer an additional clarification that may support your evaluation. **The higher number of path segments observed in the FIGR8 experiments is not an inherent limitation of the method**, but rather **a result of how the vector primitives are designed**.
> > >
> > > For example, as illustrated in the preliminary results in Figure 9, when we reconstruct the image using a layer-based approach rather than stroke-based primitives, **the number of paths required is significantly reduced**—closely resembling real-world SVG files.
> > >
> > > **SVG files generated using this layered setting are available in the same folder**, should you be interested in exploring this aspect further. [Link to folder.](https://github.com/anon-papercode/15874/tree/main/Showcase/Emoji)
> > >
> > > We hope this provides a helpful additional perspective.

---

### Official Review · Reviewer_ojwR · 2025-03-25

**Overall Recommendation:** 2

**Summary:**

This paper presents GRIMOIRE, a novel text-guided generative model for scalable vector graphics (SVG). The model consists of two main components: a Visual Shape Quantizer (VSQ), which learns to reconstruct raster images as vector shapes through a discrete codebook; and an Auto-Regressive Transformer (ART), which models the joint distribution over shape tokens, positions, and textual descriptions to generate SVGs from natural language prompts. Unlike prior approaches requiring direct supervision from SVG data, GRIMOIRE is trained only with raster image supervision, enabling it to scale to larger datasets. The authors evaluate their method on tasks such as closed-shape reconstruction (MNIST, Emoji) and stroke-based generation (icons, fonts), demonstrating improved flexibility over SVG-supervised methods and competitive generative quality against image-supervised baselines.

**Claims And Evidence:**

The proposed architecture builds incrementally upon established techniques in vector image representation and generation. It integrates standard components in a coherent manner. While the claims are plausible and generally supported by qualitative and quantitative evidence, the overall novelty is modest. The improvements are incremental, and while the results are reasonable, they do not clearly establish substantial advancement over existing methods.

**Essential References Not Discussed:**

Several key works are missing that are essential for contextualizing this paper’s contribution:
Google-Fonts (ICCV 2019): A foundational model for deep SVG generation combining image auto-encoders with SVG decoders.


DeepSVG (NeurIPS 2020): Introduced transformer-based autoencoders for vector graphics; its SVG tokenization remains widely used.


LIVE (CVPR 2022): Demonstrates SVG translation from raster images without score distillation loss.


DeepVecFont / DeepVecFont v2 (SIGGRAPH Asia 2021, CVPR 2023): Addressed SVG command modeling and differentiable rasterization.
These works are highly relevant to both the architecture and training methodology proposed in GRIMOIRE and should be discussed in the paper. Notably, vector image representation has a history that precedes the large language model (LLM) era, contrary to the implication in the related work section.

Note that representation for SVG is not starting from large language model era as mentioned in related works

**Experimental Designs Or Analyses:**

## Vector Quantization for SVG Representation (Tables 1 and 2):
 An ablation study would greatly strengthen the experimental section. For example, variations in encoder-decoder configurations (e.g., patch size, grid size, codebook size, or SVG command set) could demonstrate the robustness and contribution of the individual components. The current comparisons, such as with Im2Vec, leave some ambiguity about whether the improvements stem from the proposed method or the dataset/model choices.


## SVG Generation (Table 3):
 The evaluation should include a comparison with IconShop to give a more comprehensive view of generative quality relative to recent state-of-the-art models.

**Methods And Evaluation Criteria:**

The benchmark datasets used in this work are appropriate, as they align well with those used in prior literature, allowing for meaningful comparisons. However, the methodology would benefit from clearer descriptions of which components are used in each experiment. Since the paper combines several previously established techniques, it is important to explicitly state which variants or components are evaluated in each experiment.

**Other Comments Or Suggestions:**

## LLM Baselines:
 s a non-essential suggestion (outside rebuttal), it would be informative to include SVG generation results from public large language models (e.g., OpenAI’s GPT). Although their SVG generation quality is currently limited, showing this comparison would highlight the advantage of GRIMOIRE.


## Typos:
 There is a broken image reference at line 322:
 “Qualitative results in ?? confirm this behaviour on the MNIST dataset.”

**Other Strengths And Weaknesses:**

## Strengths:
Proposes a new framework for SVG generation that avoids the need for direct SVG supervision, opening the door to larger training datasets.
Addresses a relatively underexplored area in generative modeling, with practical applications in design, UI generation, and code synthesis.


## Weaknesses:
Lacks sufficient citation and discussion of related foundational works.
Ablation studies are missing, making it hard to isolate the contributions of each component.
Experimental gains over prior work are modest.

**Questions For Authors:**

Why use a raster-domain image encoder (e.g., ResNet-18) instead of a vector-domain encoder such as that used in DeepSVG? This choice impacts the learned latent representation, and a comparison would help clarify its effects.


How does the VSQ module differ from the VQ-based architecture in Im2Vec? The performance seems similar, so it's unclear how much the modification contributes to overall results.


Can the authors include a comparison with IconShop for SVG generation? Given that IconShop is a recent and strong baseline, its inclusion would help contextualize the performance of GRIMOIRE.

**Relation To Broader Scientific Literature:**

This work contributes to the growing field of vector image representation and generation by bridging the raster and vector domains. The approach is particularly promising for multimodal applications, including vision-language models and code generation involving SVG. Given the relevance to downstream applications and the increasing interest in multimodal generation, this research has strong potential impact.

**Theoretical Claims:**

There is no theoretical claim in this work.

---

> ### Author Rebuttal · Authors · 2025-03-30
>
> We sincerely appreciate the time and effort you have dedicated to reviewing our manuscript and providing insightful feedback.
>
> **We conducted ablation experiments to address your concerns.**
> We have explored **different patch and grid sizes** on the MNIST dataset, analyzed the **impact of stroke length and width** on FIGR8, and investigated the **effects of varying codebook sizes**. Additionally, we have included **comparisons with three vector-supervised methods**.
>
> ---
> ## Ablation: Patch and Grid Sizes (MNIST)
> We trained the VSQ module on MNIST with different grid sizes (paper value: 5) and patch sizes (paper value: 128x128).
>
> Key findings:
> - **The patch size variations had minimal impact on model performance.**
> - **Grid size variations led to improvements for larger number of patches per image**, likely due to the simpler topology of smaller patches.
> - **In all cases, the reconstruction error remains lower than Im2Vec.**
>
> The values in the table below reports the MSE on the test-set.
>
> | Patch Size | Tiles = 3 | Tiles = 5 | Tiles = 8 |
> |------------|-----------|-----------|-----------|
> | 32         | 0.093    | 0.092    | 0.078    |
> | 64         | 0.092    | 0.09      | **0.071** |
> | 128        | 0.09      | 0.094     | 0.078     |
> ---
> ## Ablation: Stroke Length (FIGR8)
> To assess the impact of stroke properties on VSQ performance, we conducted two ablations:
>
> 1. **Stroke length variations:** We created patches with smaller or larger strokes. Results show that **shorter strokes yield lower reconstruction errors**, similarly to the grid size variations.
> 2. **Multiple stroke predictions per patch:** We extended the prediction head of the VSQ to output two strokes per patch instead of one (as in the paper). Results show that more than one segment per shape consistently degrades the reconstruction quality. This suggests that the complexity of strokes in our dataset does not require multiple Bézier curves per patch.
> | Stroke Length | Segments | Stroke Width | MSE    |
> |----------------|----------|--------------|--------|
> | 3.0            | 1        | 0.4          | 0.0049 |
> | 5.0            | 1        | 0.66         | **0.011** |
> | 8.0            | 1        | 1.06         | 0.023  |
> | 3.0            | 2        | 0.4          | 0.0052 |
> | 5.0            | 2        | 0.66         | **0.017** |
> | 8.0            | 2        | 1.06         | 0.023  |
> ---
> ## Ablation: Codebook Size
> To understand the impact of codebook size $|V|$, we trained the VSQ on FIGR8 using all the sizes proposed in the original Finite Scalar Quantization paper (240, 1000, 4375 [our paper], 15360, and 64000).
>
> The results reported below highlight two key observations:
> - **The reconstruction error decreases significantly up to $|V| = 4375$ but shows only marginal improvements beyond this point.**
> - **Using excessively large codebooks does not justify the increased computational cost.**
>
> | V  | MSE |
> |-------|---------------------|
> | 240   | 0.0205             |
> | 1000  | 0.0175             |
> | 4375  | **0.0145**     |
> | 15360 | 0.0130             |
> | 64000 | 0.0128             |
> ---
> ## Comparison with Vector-Supervised Models
> **We have added a comparison with three vector-supervised models**, including a publicly available LLM: **LLama 3.2, DeepSVG, and IconShop**. We trained the models on the SVG version of FIGR8. A detailed analysis is in our response to reviewer `sRe3`.
>
> ---
> ## Missing References
> We appreciate your feedback regarding missing references.
> **We have addressed this in our response to reviewer `f1Fu` and outlined how we will incorporate these references in the final manuscript.**
>
> ---
> ## Other Questions
> > It would be helpful if the authors explicitly stated the goals of each experiment in the supplementary section.
>
> We agree. **We will add a brief introductory sentence before each subsection of the Appendix.**
>
> > Broken image reference at line 322.
>
> Thank you for catching this. We will remove the reference, as the corresponding figure is no longer part of the main manuscript.
>
> > Why use a raster-domain image encoder instead of a vector-domain encoder?
>
> A core novelty of Grimoire is that **the entire framework operates in the raster domain**.
> Using a vector-domain encoder would contradict this fundamental approach, making the pipeline **no longer vector-free**.
>
> > How does the VSQ module differ from the VQ-based architecture in Im2Vec?
>
> Im2Vec utilizes an **RNN-based architecture**, which does not belong to the family of **VQ models using discrete embedding codebooks**.
>
> Another key methodological difference is that Im2Vec attempts **end-to-end SVG generation**, encoding an entire image and predicting the individual components with an RNN. In contrast, our VSQ first learns vector representations from image patches, then the ART model learns to arrange these patches in the correct sequence.
>
> This modular approach significantly **enhances the performance of raster-supervised generative models** compared to Im2Vec.

---

### Decision · Program_Chairs · 2025-05-01

**Decision:**

Accept (poster)

**Comment:**

This paper received 2 “Weak reject” and 2 ”Weak accept.” It is a borderline paper that needs further discussion with the SAC to make the final decision. Currently, the AC is slightly leaning towards rejecting this paper based on the analyses of its strengths and weaknesses listed below.

Strengths:
1. A new text-guided SVG generation model was proposed using only raster image supervision.
2. The idea of splitting the image to patches in VSQ stage is interesting.

Weaknesses:
1. The technical contribution and the performance improvements over existing methods are incremental.
2. Many important related works (e.g., Google-Fonts, DeepSVG, LIVE, DeepVecFont, etc.) are missing, which should be cited, discussed and compared.
3. Insufficent experiments have been conducted.
4. Although the proposed method can be trained using only raster image supervision, it cannot outperform some approaches trained on ground-truth paths. Therefore, it fails to handle the practical real-world scenario where the model is trained on vector graphs and outputs high-quality SVGs.